# Kinetic and structural roles for the surface in guiding SAS-6 self-assembly to direct centriole architecture

Niccolò Banterle [1,8], Adrian P. Nievergelt [2,6,8], Svenja de Buhr[3,4], Georgios N. Hatzopoulos [1], Charlène Brillard [2], Santiago Andany[2], Tania Hübscher[1], Frieda A. Sorgenfrei [4,7], Ulrich S. Schwarz [3,5], Frauke Gräter [3,4], Georg E. Fantner [2] & Pierre Gönczy [1✉]

Discovering mechanisms governing organelle assembly is a fundamental pursuit in biology. The centriole is an evolutionarily conserved organelle with a signature 9-fold symmetrical chiral arrangement of microtubules imparted onto the cilium it templates. The first structure in nascent centrioles is a cartwheel, which comprises stacked 9-fold symmetrical SAS-6 ring polymers emerging orthogonal to a surface surrounding each resident centriole. The mechanisms through which SAS-6 polymerization ensures centriole organelle architecture remain elusive. We deploy photothermally-actuated off-resonance tapping high-speed atomic force microscopy to decipher surface SAS-6 self-assembly mechanisms. We show that the surface shifts the reaction equilibrium by ~$10^4$ compared to solution. Moreover, coarse-grained molecular dynamics and atomic force microscopy reveal that the surface converts the inherent helical propensity of SAS-6 polymers into 9-fold rings with residual asymmetry, which may guide ring stacking and impart chiral features to centrioles and cilia. Overall, our work reveals fundamental design principles governing centriole assembly.

[1] Swiss Institute for Experimental Cancer Research (ISREC), School of Life Sciences, Swiss Federal Institute of Technology Lausanne (EPFL), CH-1015 Lausanne, Switzerland. [2] Laboratory for Bio- and Nano-Instrumentation, Swiss Federal Institute of Technology Lausanne (EPFL), CH-1015 Lausanne, Switzerland. [3] Interdisciplinary Center for Scientific Computing (IWR) Heidelberg University, D-69120 Heidelberg, Germany. [4] Heidelberg Institute for Theoretical Studies, Heidelberg, Germany, D-69118 Heidelberg, Germany. [5] Heidelberg University, Institute for Theoretical Physics and BioQuant, D-69120 Heidelberg, Germany. [6] Present address: Max Planck Institute of Molecular Cell Biology and Genetics, Pfotenhauerstraße 108, 01307 Dresden, Germany. [7] Present address: Austrian Centre of Industrial Biotechnology c/o University of Graz, Institute of Chemistry, NAWI Graz, BioTechMed Graz, Heinrichstrasse 28, 8010 Graz, Austria. [8] These authors contributed equally: Niccolò Banterle, Adrian P. Nievergelt. ✉email: pierre.gonczy@epfl.ch

Deciphering the mechanisms directing organelle biogenesis is paramount for understanding cell physiology and pathological conditions. The centriole organelle is particularly attractive for achieving this goal because many of the molecular components are identified, and their assembly results in a signature architecture that is crucial for fundamental cellular processes, such as polarity, division and motility (reviewed in refs. [1,2]). Abnormalities in centriole number and architecture are associated with several human diseases, including ciliopathies and cancer (reviewed in refs. [3–5]). Despite their importance, the biophysical principles guiding the onset of centriole biogenesis remain elusive, limiting understanding of how proper centriole architecture is organized and may go awry in disease conditions.

The centriole is typically ~500 nm (diameter) × 250 nm (height) in dimensions and exhibits a characteristic and evolutionary conserved 9-fold radial symmetry of microtubules. In addition, the arrangement of microtubule triplets exhibits chirality, in that microtubules are tilted clockwise when viewed from the distal end of the organelle (Fig. 1a, inset, reviewed in ref. [6]). This geometry is thought to be critical for the function of the motile cilia and flagella that are seeded by the centriole, and which exhibit likewise 9-fold symmetrical and chiral features. In the canonical centriole duplication cycle, one centriole forms near-orthogonal with respect to the long axis of each resident centriole. The first detectable structure in the nascent centriole is the cartwheel, which comprises stacked 9-fold symmetrical ring polymers, with a central hub ~22 nm in diameter from which

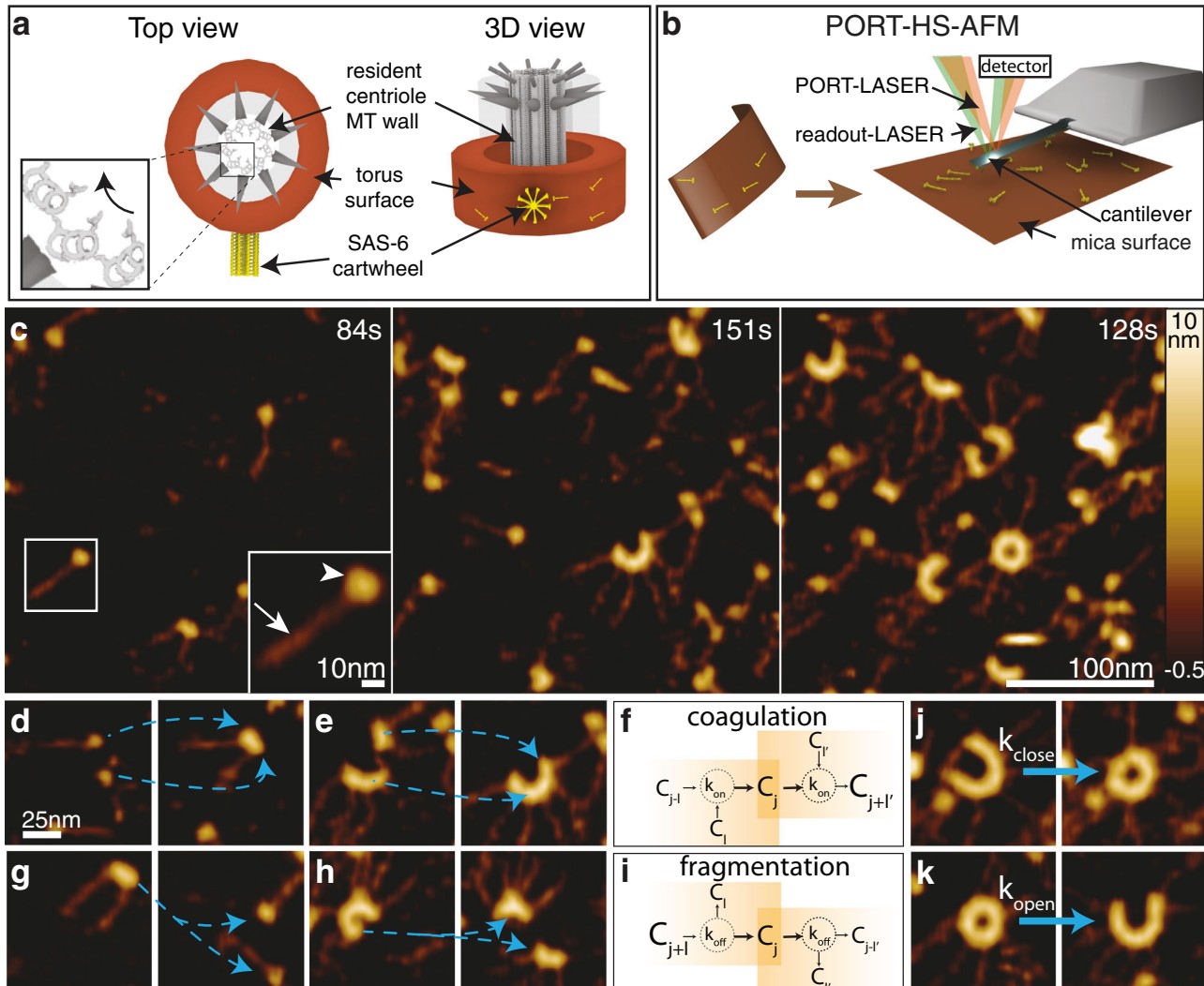

**Fig. 1 PORT-HS-AFM of surface guided SAS-6 self-assembly exhibits characteristics of coagulation-fragmentation system. a** Schematic top view (left, viewed from the distal end) and 3D front view (right) of centriole with chiral arrangement of microtubule (MT) wall (bottom left inset, see curved arrow) surrounded by toroidal surface (brown) and cartwheel formed by SAS-6 homodimers (yellow). Note that the cartwheel emerges orthogonal to the resident centriole, including in *Chlamydomonas*[67]. **b** Illustration of correspondence between torus surface and mica surface in PORT-HS-AFM setting. In PORT-HS-AFM, SAS-6 homodimers are adsorbed on the mica surface where their shape and position are probed by the cantilever, the motion of which is actuated by the PORT-LASER, with height monitored by the readout-LASER. **c** Frames from SAS-6 self-assembly reaction monitored with PORT-HS-AFM. Higher magnification inset at 84s shows single homodimer with head domain (arrowhead) and spokes (arrow). Here and hereafter: time relative to start of recording. See also Supplementary Movie 1 (for **c**–**e**, **g**, **h**). **d**, **e** Association of SAS-6 homodimers monitored by PORT-HS-AFM (**d**: 1-mer+1-mer = 2-mer, **e**: 4-mer+2-mer = 6-mer). Here, as well as in **g**, **h**, **j** and **k**, pairs of frames are 2.52s apart and transitions are marked by blue dashed arrows. **f** Coagulation reactions can lead to either appearance or disappearance of an intermediate j-mer. Here and in **i**, see Eq. (1) in text and Supplementary Note 1 for details. **g**, **h** Dissociation of SAS-6 homodimers monitored by PORT-HS-AFM (**g**: 2-mer = 1-mer+1-mer, **h**: 7-mer = 4-mer+3-mer). **i** Fragmentation reactions can lead to either appearance or disappearance of an intermediate j-mer. **j**, **k** SAS-6 ring closing (**j**) and opening (**k**) events.

emanate ~40 nm-long spokes that connect through a pinhead element to peripheral microtubules (reviewed in ref. [7]). How the biophysical properties of individual building blocks dictate the geometrical properties of the nascent organelle, including orthogonal emergence and chiral features, remains to be discovered.

The main constituent of the ring polymers present in the cartwheel is the evolutionarily conserved SAS-6 protein family. SAS-6 proteins harbor a globular N-terminal head domain followed by a coiled-coil domain and an unstructured C-terminal moiety[8,9]. Through their coiled coil, SAS-6 proteins form homodimers that can undergo higher order oligomerization to form a 9-fold symmetrical ring polymer akin to the structure observed in the cellular context[8,9]. In human cells, HsSAS-6 is recruited to a toroidal surface that surrounds the resident centriole, and which contains the interacting proteins Cep152/Cep63/Cep57 (reviewed in ref. [10]). In the prevailing working model, HsSAS-6 is recruited to a specific location on this torus defined by a focus of the Plk4 kinase and its substrate STIL. Because STIL also binds HsSAS-6, such focusing could direct HsSAS-6 recruitment and assembly to a single location, thus ensuring formation of only one nascent centriole per resident centriole (reviewed in ref. [11]). An analogous situation is encountered in *Drosophila*, where DmSAS-6 emerges from a torus containing the Cep152-related protein Asl, in a manner that depends on Plk4 and the STIL homolog Ana2[12]. Upon recruitment to the torus, SAS-6 proteins are thought to somehow form ring polymers and then stack in the vertical direction, orthogonal to the resident centriole, thereby generating a structural scaffold for the nascent centriole organelle (reviewed in refs. [7,13]). While the prevailing view is that SAS-6 polymerization occurs at the focus containing Plk4 and phosphorylated STIL, some observations appear to question this view. For instance, overexpression of SAS-6 proteins results in supernumerary centrioles in both human cells and *Drosophila*[14,15], suggesting that excess SAS-6 can bypass or alter the restriction imposed by the singularity of the focus of Plk4 and phosphorylated STIL. Moreover, a HsSAS-6 mutant defective in ring polymer formation localizes throughout the torus[16], indicating that focusing of SAS-6 proteins requires their polymerization.

SAS-6 homodimers interact with one another through their N-terminal head domain, with a dissociation constant ($K_d$) of ~60 μM in solution[8,9]. For the human protein, this is ~1000 times higher than the cytoplasmic concentration[16,17], making it unclear how SAS-6 proteins could self-assemble into ring polymers in the cellular context. One possibility is that post-translational modifications or partner proteins alter the effective $K_d$ for SAS-6 self-assembly in cells. Another possibility is that SAS-6 proteins self-assemble following concentration through liquid-liquid phase separation (LLPS), which has been proposed to operate in the case of Plk4[18–20]. Alternatively, theoretical considerations raised the possibility that the presence of a surface might promote SAS-6 self-assembly[21], but whether this is the case has not been addressed experimentally.

To understand how the fundamental SAS-6 building block can self-assemble into ring polymers with the correct symmetry and guide proper organelle architecture, it is necessary to quantitatively probe the properties of the SAS-6 surface polymerization reaction. Cell free assays have been transformative for revealing the fundamental features of other self-assembling cellular polymers, including microtubules, F-actin and FtsZ rings (reviewed in refs. [22–24]). Understanding such fundamentals of polymer dynamics has proven critical not only for uncovering the self-organizing properties of the corresponding cytoskeletal networks, but also for revealing how these properties can be harnessed and modulated in the cellular context (reviewed in ref. [25]).

By analogy, unraveling the fundamental surface polymerization properties underlying SAS-6 ring assembly is expected to shed critical light on the structural principles governing biogenesis of centriole architecture.

Here, we show that surface SAS-6 ring polymer assembly is an aggregation-fragmentation process in which dynamic ring opening and closing is critical to favor 9-fold symmetrical arrangements. We uncover a dual role for the surface in guiding SAS-6 assembly: the surface shifts the effective equilibrium $K_d$ by ~$10^4$ compared to solution and promotes assembly of planar SAS-6 ring polymers. We find also that SAS-6 ring polymers retain residual asymmetry from their intrinsic helical propensity, leading us to propose a new mechanism of cartwheel assembly that may impart chirality to the centriole.

## Results

**HS-PORT-AFM reveals dynamicity and reversibility of SAS-6 self-assembly.** High-speed atomic force microscopy (AFM) is a powerful experimental approach to probe surface protein dynamics at the nanometer scale (reviewed in ref. [26]). We previously developed photothermally-actuated off-resonance tapping high-speed AFM (PORT-HS-AFM) as a minimally invasive method to monitor surface polymerization reactions[27]. Here, we set out to deploy PORT-HS-AFM to achieve a quantitative understanding of surface-guided SAS-6 self-assembly, and thus unveil the design principles guiding the onset of centriole biogenesis (Fig. 1b). This analysis was conducted with the *Chlamydomonas reinhardtii* protein SAS-6[NL] (hereafter referred to for simplicity as SAS-6, Supplementary Fig. 1), which contains the globular N-terminal head domain and the coiled-coil corresponding to the spokes, but not the unstructured C-terminal moiety[9]. We determined the monomer-dimer equilibrium constant ($K_d$) of the purified protein to be ~20 ± 15 nM (Supplementary Fig. 1), such that SAS-6 readily formed homodimers in solution at the concentration of ~120 nM employed in these experiments (see PORT-HS-AFM imaging section in Methods). Given that the $K_d$ for interaction between head domains is ~60 μM[9], higher order oligomers are not expected to form in solution at this concentration. We found that SAS-6 homodimers progressively adsorbed on the charged mica surface, where PORT-HS-AFM detected SAS-6 homodimers with their characteristic head domains from which spokes emanate (Fig. 1c, inset). We found that SAS-6 homodimers diffused on the surface and then bound to each other through their head domains, leading over time to the formation of higher order oligomers and eventually rings (Fig. 1c, Supplementary Movie 1)[27].

Initial analysis of such assembly reactions revealed several important features. First, oligomerization, which can also be referred to as a coagulation event, occurred both through the incorporation of single homodimers onto pre-existing oligomers (Fig. 1d), and the association of higher order oligomers (Fig. 1e, f). Second, dissociation or fragmentation events likewise occurred both for homodimer pairs and higher order oligomers (Fig. 1g–i). These findings demonstrate that the SAS-6 self-assembly reaction is reversible. Third, we observed that polymers containing 7 to 10 homodimers in an open configuration could each transition into closed rings (Fig. 1j shows such a transition for a 9-fold polymer). Interestingly, we found that ring formation is also reversible, since closed rings could open into curved polymers without homodimer loss (Fig. 1k). Together, these findings indicate that surface-guided SAS-6 self-assembly is a complex dynamic system. A thorough understanding of the underlying properties of such a system requires a quantitative model of the self-assembly reaction.

**SAS-6 assembly is described by a coagulation-fragmentation process**. We reasoned that a coupled differential equation system describing coagulation-fragmentation processes provided an appropriate mathematical framework for the SAS-6 self-assembly reaction. Indeed, such equations are well suited to model reversible growth processes in physical and biological systems (reviewed in refs. [28,29]). Therefore, a coagulation-fragmentation equation system seemed well suited for SAS-6 self-assembly, considering the observed reversible reactions between higher order oligomers, as well as between ring opening and closing events. However, because SAS-6 forms ring polymers, such equations must be modified to account for the presence of open and closed rings (Supplementary Note 1)[21]. Therefore, we developed equations comprising the sum of six terms: (a) two terms for the appearance and disappearance of a j-mer due to coagulation (Fig. 1f), with j-mers comprising 1–10 homodimers; (b) two terms for the appearance and disappearance of a j-mer due to fragmentation (Fig. 1i); (c) two terms for ring closing and ring opening for j-fold rings comprising 7–10 homodimers. We assumed identical association ($k_{on}$) and dissociation ($k_{off}$) rate constants regardless of oligomer size, as expected in a reaction-controlled system (Supplementary Note 1). The resulting coupled differential equations read:

$$\frac{dC_j}{dt} = k_{on}\sum_{l=1}^{j-1}C_l \cdot C_{j-l} - 2k_{on} \cdot C_j\sum_{l=1}^{10-j}C_l + 2k_{off}\sum_{l=j+1}^{10}C_l$$
$$- k_{off} \cdot (j-1) \cdot C_j - k_{close}^{j}\sum_{l=7}^{10}\left(\delta_{j,l} \cdot \bar{C}_j\right) \quad (1)$$
$$+ k_{open}^{j}\sum_{l=7}^{10}\left(\delta_{j,l} \cdot \bar{C}_j\right)$$

where $C_j$ is the concentration of j-mers and $\bar{C}_j$ the concentration of closed j-fold rings; $k_{close}^{j}$ and $k_{open}^{j}$ correspond to ring closing and opening rates, respectively, which can depend on the symmetry type j of the corresponding ring and δ is the Kronecker delta function.

**Quantification of SAS-6 assembly dynamics**. We set out to determine the parameters of this coagulation-fragmentation model. To uncover the dissociation rate $k_{off}$, we utilized experimental conditions where only a few homodimers were present on the surface, such that individual dissociation events were readily quantifiable (Fig. 2a) (see PORT-HS-AFM imaging section in Methods). We found that the distribution of SAS-6 homodimer-homodimer dissociation time on the surface yielded a decay constant of 0.046 ± 0.005 Hz (Fig. 2b).

We proceeded to determine the opening and closing rate as a function of ring symmetry to determine $k_{open}^{j}$ and $k_{close}^{j}$ in each case. Ring polymers containing 7 to 10 homodimers were pre-formed on mica (Fig. 2c–f), and the rates of spontaneous opening and closing were monitored for those rings that did not undergo homodimer addition or loss during the experiment (Supplementary Movie 2). We thus determined for how long a newly closed ring of symmetry j remained in that state. We found that the lifetime of 7-fold_closed rings was much shorter than that of other symmetries, accounting for their very transient nature (Fig. 2g, i, Supplementary Fig. 1). We also measured for how long a newly opened ring of symmetry j remained in that state (Fig. 2h, Supplementary Fig. 1). Interestingly, we found that the average lifetime of 9-fold_open rings was significantly shorter than that of 8-fold_open rings, meaning that 9-fold open polymers closed sooner (Fig. 2j). Together, the symmetry-dependent kinetic differences in closing and opening lifetimes lead to a stabilization of rings with

9-fold symmetry, potentially helping explain why 9-fold rings represent the majority equilibrium population[30].

Having determined $k_{off}$ for homodimer-homodimer interactions, as well as $k_{close}^{j}$ and $k_{open}^{j}$ for ring polymers containing 7 to 10 homodimers, we set out to estimate $k_{on}$ by fitting the coagulation-fragmentation equation system (1) to the experimental data, and thus compute the effective surface dissociation constant $K_d = \frac{k_{off}}{k_{on}}$. We first determined the frequency of individual oligomeric species during each step of the assembly reaction by segmenting the PORT-HS-AFM data using machine learning aided classification[31]. We trained a model to classify the different oligomeric assemblies that had been imaged, placing each of them into one of 14 classes: 1–10 SAS-6 homodimers, plus 4 classes for the 7–10-fold closed rings (Fig. 3a, b, Supplementary Fig. 2). The accuracy of the automatic classification was validated by comparing it with manual classification performed on a subset of the data independently by two of the authors (Supplementary Fig. 2). A total of 38,539 oligomeric assemblies were automatically classified in five self-assembly reactions, leading to the determination of the abundance of each type of oligomeric species over time (Fig. 3c, Supplementary Fig. 3). As can be seen by the slope in Fig. 3d, we observed a stereotyped kinetics of the self-assembly reaction upon normalizing the data as a function of homodimers surface concentration. Taken together, these experiments enabled us to determine critical parameters of the coagulation-fragmentation model and unveil the kinetics of surface-guided SAS-6 self-assembly.

**The surface shifts the equilibrium $K_d$ by ~$10^4$ compared to solution**. We were thus in a position to fit the experimental data with the aggregation-fragmentation equation system (1) (Fig. 4a, Supplementary Fig. 3), assessing validity with a goodness of fit test and optimality versus simpler or more complex models (Supplementary Fig. 3). The fitting yielded an average effective surface dissociation constant $K_d$ of ~79 ± 12 homodimers/μm². Considering that we determined the average surface concentration to be $C_{surf} \sim 500$ homodimers/μm² (Supplementary Fig. 3; see also Kinetics fitting procedure in Methods), the ratio between the two surface values is $R_{surf} = \frac{C_{surf}}{K_d^{surf}} = \frac{500}{79} \sim 10$. How does this compare to the situation in solution? The concentration of SAS-6 homodimers in solution in these experiments is $C_{sol} \sim 120nM$, whereas the dissociation constant in solution is $K_d^{sol} \sim 60\mu M$[9], leading to a ratio between the two solution values of $R_{sol} = \frac{C_{sol}}{K_d^{sol}} = \frac{0.12}{60} \sim 10^{-3}$. Therefore, the surface confers a catalytic effect that shifts the equilibrium by a factor ~$10^4$ towards higher order oligomerization.

Could an effective $K_d$ of 79 ± 12 homodimers/μm² enable self-assembly in the cellular context? A definite answer to this question cannot be provided at this stage, given that mica contributes an interaction surface that may have distinct properties from those of the torus present in cells; moreover, the mica surface is flat, in contrast to the curved cellular toroidal surface. These considerations notwithstanding, we note that approximately 100 HsSAS-6 homodimers have been estimated to be present at centrioles by quantitative mass-spectrometry[17], presumably reflecting proteins in the cartwheel, which were likely located on the torus prior to stacking. Given that the surface of the torus is ~0.1 μm²[32], this would translate into an approximate effective surface concentration of ~1000 homodimers/μm². This is 10 times higher than the $K_d$ determined here, potentially providing a mechanism for why SAS-6 rings self-assemble exclusively on the surface of the torus and not in the cytosol in the cellular context. Overall, we conclude that surface-guided

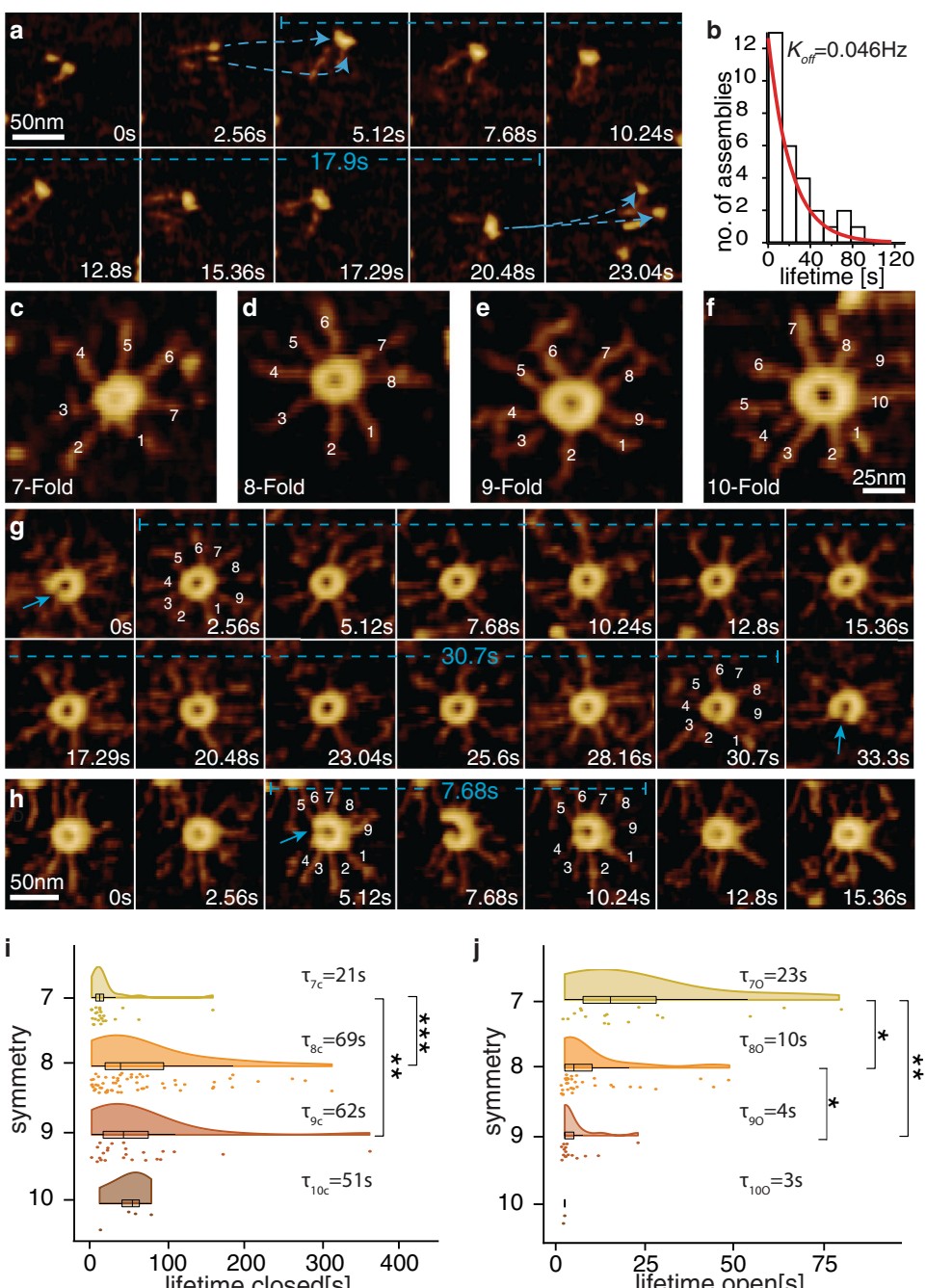

**Fig. 2 Determination of $k_{off}$, $k^j_{close}$ and $k^j_{open}$ for surface SAS-6 self-assembly. a** Lifetime (17.9 s, dashed blue line) of a 2-mer analyzed with PORT-HS-AFM, forming at 5.12 s and dissociating at 20.48 s (dashed blue arrows indicate association and dissociation). **b** Distribution of lifetimes of 2-mer, with exponential fitting (red line), yielding $k_{off} = 0.046 \pm 0.005$ Hz; $n = 30$. **c–f** Representative images of SAS-6 rings of indicated symmetries (determined by the number of spokes) from $N = 24$ movies observed by PORT-HS-AFM. Note that ring diameter increases with higher fold-symmetries. **g**, **h** Lifetime of closed (**g**) and open (**h**) SAS-6 rings analyzed with PORT-HS-AFM; blue arrows indicate opening events, dashed blue lines lifetimes. See also Supplementary Movie 2. **i**, **j** Distribution of lifetime of closed (**i**) and open (**j**) rings with indicated symmetries and corresponding average lifetimes; The box shows the upper (75%) and lower (25%) quartiles and mean, the whiskers represent upper quartile +1.5*IQR (inter quartile range) and lower quartile −1.5*IQR, respectively. Asterisks (*) indicate statistically significant differences (F-ratio test on the means; see Supplementary Table 1 for p-values and number of transitions measured for each condition).

catalysis surmounts the weak interaction and low concentration that homodimers exhibit in solution, thus enabling efficient in vitro SAS-6 self-assembly.

We used the above quantitative model to computationally explore the impact of changing the effective surface concentration and the dissociation constant between homodimers on the efficiency and the symmetry of ring formation. We thus

uncovered that ring formation efficiency exhibited a strong dependence on effective surface concentration and dissociation constant (Fig. 4b). In addition, the average ring symmetry at equilibrium was remarkably robust to parameter changes, remaining close to 9-fold within the entire explored range (Fig. 4c). Furthermore, we found that symmetries remained essentially unchanged when lifetimes of closed rings became

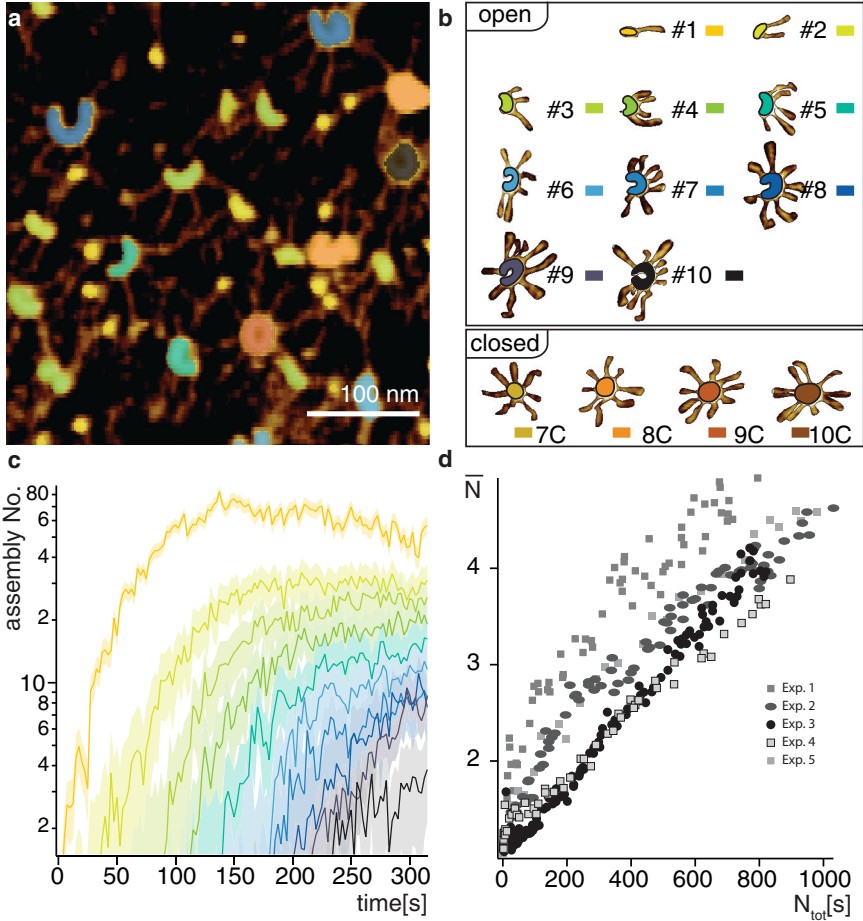

**Fig. 3 Machine learning aided classification of surface SAS-6 self-assembly. a** Distinct oligomeric states recognized by machine learning aided classification in PORT-HS-AFM data. **b** Representative examples of existing classes. In total there are 10 open and 4 closed conformations, some of which are present in the image shown in panel **a**. **c** Number of assemblies of each oligomeric species over time (thin lines, colors as in panel B) in one PORT-HS-AFM experiment. **d** Average oligomerization degree ($\overline{N}$) as a function of the total number of homodimers on the mica surface for 5 experiments (see Supplementary Fig. 3 for individual time traces). Note similar slopes, reflecting analogous oligomerization kinetics.

shorter, which mimics enhanced reversibility (Fig. 4d right-most). By contrast, when lifetimes of closed rings became longer, which mimics poor reversibility, symmetry shifted towards 7-fold due to kinetic trapping (Fig. 4d, left-most). Overall, we conclude that symmetry-dependent lifetimes of open and closed rings, as well as reversibility of the assembly reaction, together ensure proper 9-fold ring symmetry over a broad parameter range.

**The surface converts helical SAS-6 polymers into 9-fold ring polymers**. What are the physical reasons underlying the observed symmetry-dependent open and closed ring lifetimes? The fundamental parameters dictating opening and closing of any ring are the native curvature and flexibility of its constituent polymer (reviewed in[33]). To evaluate these parameters in the case of SAS-6, we analyzed 4- and 6-mer intermediates, in which curvature is not influenced by closing constraints, since reactive ends are far from one another (Fig. 5a, Supplementary Fig. 4). As shown in Fig. 5a, we found that the curvature, as expressed by the mean angle between neighboring SAS-6 homodimers in 4- and 6-mers, is $\alpha_4 = 39° \pm 4°$ and $\alpha_6 = 38° \pm 3°$, respectively. These values reflect a curvature that would yield rings with a symmetry between 9- and 10-fold, with polymer flexibility reflected by the standard deviation (Fig. 5a).

Interestingly, computational assembly of the SAS-6 homodimer with 6 heptad repeats of the coiled-coil (the longest

available crystal structure, hereafter referred to as SAS-6[6HR]; Supplementary Fig. 1a) shows that whereas 9 SAS-6[6HR] homodimers yielded a flat ring when enforced by modeling (Fig. 5b, blue), in the absence of any constraints it yielded instead a shallow helix with a ~7.9 nm pitch (Fig. 5b, red)[9,34]. What are the consequences of such a helical propensity for the nature of the ring assembled on the surface? From a purely energetic point of view, the final shape of a ring assembling on a surface is equivalent to that of the helix constrained on a plane. A theory of the preferred conformation of helices when constrained on surfaces predicts the existence of metastable states, including a signature S-shaped conformation[35,36]. Remarkably, such transient strained SAS-6 conformations were indeed present in the PORT-HS-AFM data set (Fig. 5c–f, Supplementary Movie 3). Taken together, these findings support the notion that SAS-6 polymers exhibit an intrinsic helical propensity that is of consequence for ring polymers assembled on the surface.

To gain quantitative molecular insight into ring polymers formed on the surface by an intrinsically helical SAS-6 polymer, we performed extensive coarse-grained molecular dynamics (MD) computer simulations of SAS-6[6HR] in different oligomerization states, both in solution and on a surface. The resulting distribution of angles between homodimers in the plane of the ring revealed that the conformational ensemble for 4- and 6-mers in solution would result in a ~6-fold symmetry (Fig. 5g, Supplementary Fig. 4). Strikingly, we found that the presence of

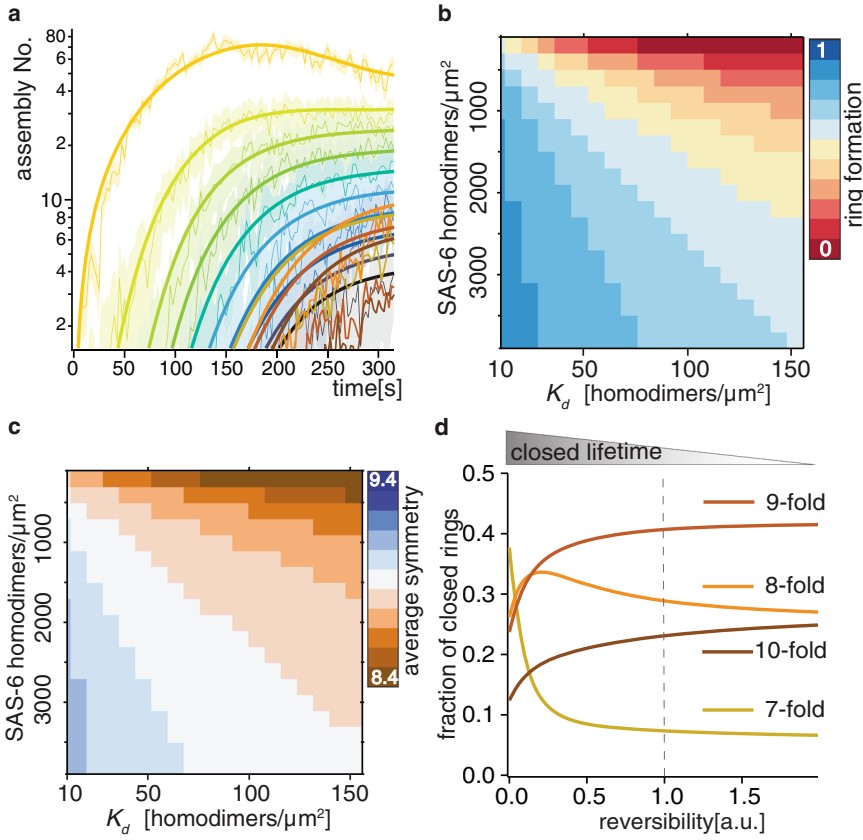

**Fig. 4 Modeling of surface SAS-6 self-assembly. a** Fit with coagulation-fragmentation model (overlaid thick lines) of the experimental evolution over time of SAS-6 assembly reaction from Fig. 3c. **b, c** Model predictions of average ring formation efficiency (**b**) and average ring symmetry (**c**) at varying $K_d$ and SAS-6 homodimers/$\mu m^2$. **d** Fraction of closed rings of indicated symmetries calculated as numerical solution of the fitted model after 17 min as a function of closed lifetime (1 = experimental values, <1 slower opening, >1 faster opening).

a surface significantly decreased such in-plane angles, bringing them close to a configuration enabling 9-fold symmetry (Fig. 5g, Supplementary Fig. 4), in agreement with the values determined experimentally with PORT-HS-AFM (see Fig. 5a). We conclude that the surface plays a critical role in promoting in-plane angles close to 40° between neighboring SAS-6 homodimers, thus favoring assembly of 9-fold symmetrical ring polymers.

Another remarkable feature unveiled by the MD simulations is that SAS-6[6HR] rings were not entirely flat on the modeled surface, since spokes were either bound or not bound to it (Fig. 5h), as evident from the torsion angle η (Fig. 5i, Supplementary Fig. 4). This was the case also when surface-protein and surface-water interactions parameters were varied in the MD simulations (Supplementary Fig. 5a–c). Furthermore, statistical analysis revealed that the spatial distribution within each ring of spokes being either bound or not bound was not random for most parameter sets (Fig. 5j, Supplementary Fig. 5d). This distribution was random only for parameters sets in which the fraction of bound spokes was >60%, where it becomes inevitably difficult to detect a non-random distribution (Supplementary Fig. 5d). Overall, these simulations suggest that the intrinsic helical propensity of the SAS-6 polymer results in rings that are not perfectly planar on the surface, with spokes exhibiting flexibility in the vertical direction. Interestingly, such breaking of planar symmetry also breaks the radial symmetry of SAS-6 rings, since an odd number of spokes cannot be divided into an even set of bound and not bound components.

Given that MD simulations showed that bound spokes are less mobile on the surface than those that are not bound (Supplementary Fig. 4h), we used spoke mobility as a proxy for surface interaction in the PORT-HS-AFM data set, finding that individual spokes indeed have different motilities (Supplementary Fig. 5j–n). Overall, we conclude that the intrinsic helical propensity of the SAS-6 polymer translates into a residual asymmetry on the surface that breaks the planar and radial symmetries of the ring polymer.

**Residual asymmetry of SAS-6 ring polymers may guide their stacking.** Albeit needed for ring formation, the interactions brought about by the surface are likely to dampen a potential pattern of spoke orientation along the ring circumference. Therefore, we reasoned that the pattern of spoke orientations along a virtual ring closing in solution could fully uncover their intrinsic conformational properties. Moreover, we wondered whether such a putative pattern might have fundamental consequences on the configuration of stacked rings. Hence, we conducted MD simulations of virtual closed rings in solution (Fig. 6a). Interestingly, this revealed an oscillatory behavior of the torsion angle η as a function of circumferential spoke position (Fig. 6b). We reasoned that such an oscillatory feature, if remaining in the ring assembled on the surface, could impact the geometry of the stack of rings by favoring contacts between spokes of consecutive rings based on their minimal energy configuration. This could result in a final stacked configuration where only some spokes of two consecutive rings are in contact (Fig. 6c, arrowheads), leaving the remaining spokes available for interaction with spokes from the next ring (Fig. 6c). This scenario is different from the current stacking model, which is based on cartwheel reconstructions from averaged and symmetrized cryo-electron tomograms (cryo-ET)[37,38]. In that model, SAS-6 rings

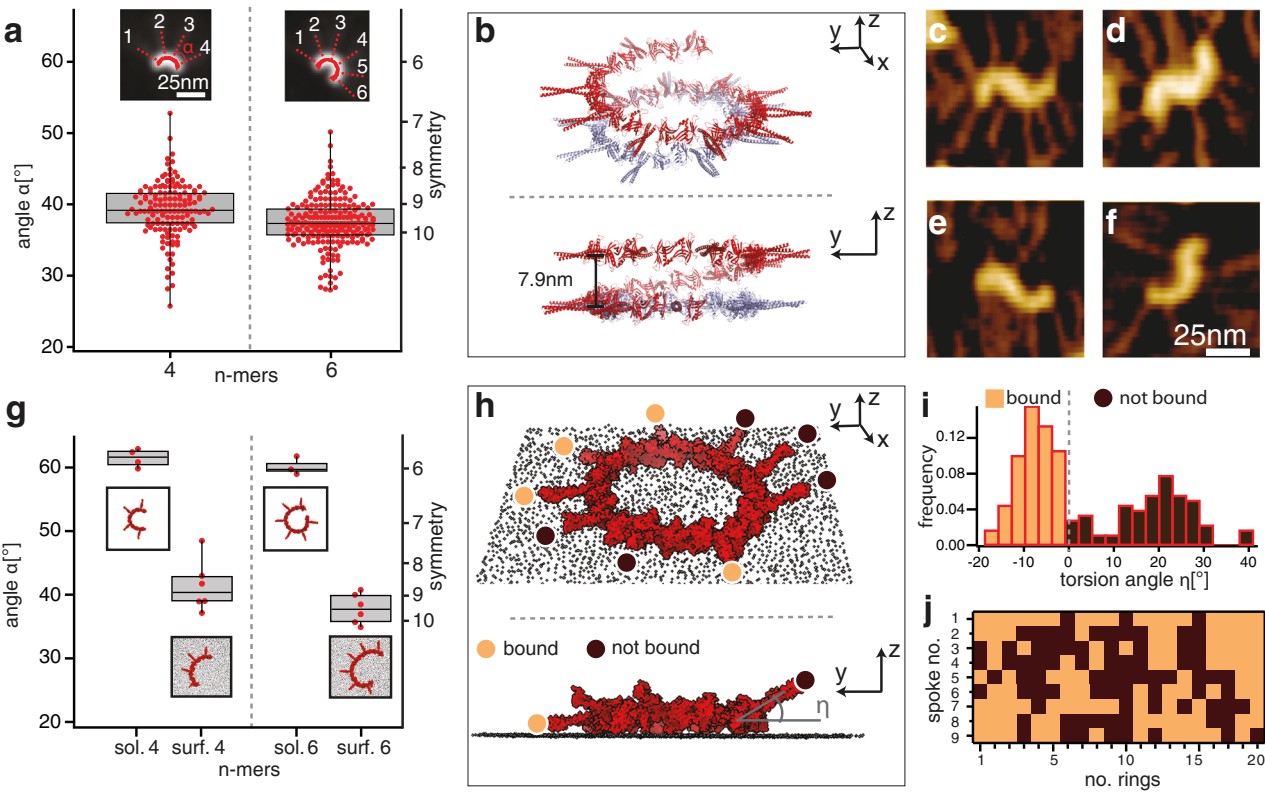

**Fig. 5 Surface-guided symmetry of the SAS-6 ring polymer. a** Box plot of angle ($\alpha$) distribution of 4- and 6-mers of SAS-6 from PORT-HS-AFM data set (mean ± SD: $\alpha_4 = 39° ± 4°$, N = 144; $\alpha_6 = 38° ± 3°$, N = 215). The box shows the upper and lower limits as calculated by the Tukey method and mean, the whiskers extend from the minimum to the maximum of the experimental data. Insets: averaged and realigned images with fit (solid red line, dashed red lines denote spoke locations). **b** Isometric (top) and side (bottom) views of 9-mer generated from SAS-6[6HR] crystal structures resulting in a spiral (red); the closed ring (blue) is enforced by modeling[9]. **c–f** Examples of transient S-shaped SAS-6 oligomers observed with PORT-HS-AFM in N = 5 independent experiments. See also Supplementary Movie 3. **g** Box plot of in-plane ($\alpha$) angle distribution from MD simulations of SAS-6[HR] 4- and 6-mers in solution and on a surface (the number of simulations for each condition is: for sol 4. N = 4, for surf 4. N = 6, for sol 6 N = 3, for surf 6 N = 6). Insets show examples of corresponding simulated structures. The box shows the upper (75%) and lower (25%) quartiles and median, the whiskers represent upper quartile +1.5*IQR (inter quartile range) and lower quartile −1.5*IQR, respectively. **h** Isometric (top) and side (bottom) view from MD simulation of closed SAS-6[HR] 9-mer on a surface, with spokes either bound to the surface (beige disks) or not bound (brown disks); only two spokes' directions are highlighted in this fashion in the side view. **i, j** Torsion angle ($\eta$) distribution of 20 closed SAS-6[HR] 9-mers on a surface simulated by MD (**i**), and corresponding matrix representation (**j**), with spokes classified as bound or not bound (bound when torsion angle <0, dashed line in **i**).

are stacked two by two, with the corresponding spokes merging pair-wise towards the periphery (Supplementary Fig. 6a–d). In such a paired model, all spokes within one ring are predicted to be oriented in the same manner, with a constant torsion angle $\eta$ (Supplementary Fig. 6). To test whether this may be the case, we extracted individual cartwheels from an extant cryo-ET data set of in vitro assembled SAS-6 stacks[39], now analyzing them without symmetrization or sub-tomogram averaging (Fig. 6d). After image filtering, we focused on individual sectors that each correspond to a stack of spokes (Fig. 6e, f) (see Tomogram processing section in Methods). Importantly, this analysis revealed that not all spokes within each ring have the same angle $\eta$, as evidenced by spokes in adjacent sectors not being parallel to one another (Fig. 6f), thus falsifying the prediction from the paired model.

Therefore, we generated a geometrical model of stacked rings with spoke orientations drawn from the MD simulations, being fixed in the vertical direction and oscillating around the ring circumference. Note that this oscillating model pertains to the end state and does not inform on the dynamics or the polarity of the stacking reaction. In addition, the model assumes that the relative position between consecutive rings is determined by maximizing spoke contact with the previous ring (Fig. 6g) (see Geometrical stacking modeling section in Methods). We found that the resulting 3D arrangement exhibits adjacent rings that are not in phase (Fig. 6h and Supplementary Fig. 6), which is compatible with the experimental data (see Fig. 6f). Interestingly this arrangement does not exhibit radial or mirror symmetries (Supplementary Fig. 6). Therefore, it is chiral in nature, as can be also seen in the unwrapped side view shown in Fig. 6i.

### Discussion
Deciphering the mechanisms governing centriole assembly has been a fundamental goal in cell and developmental biology since

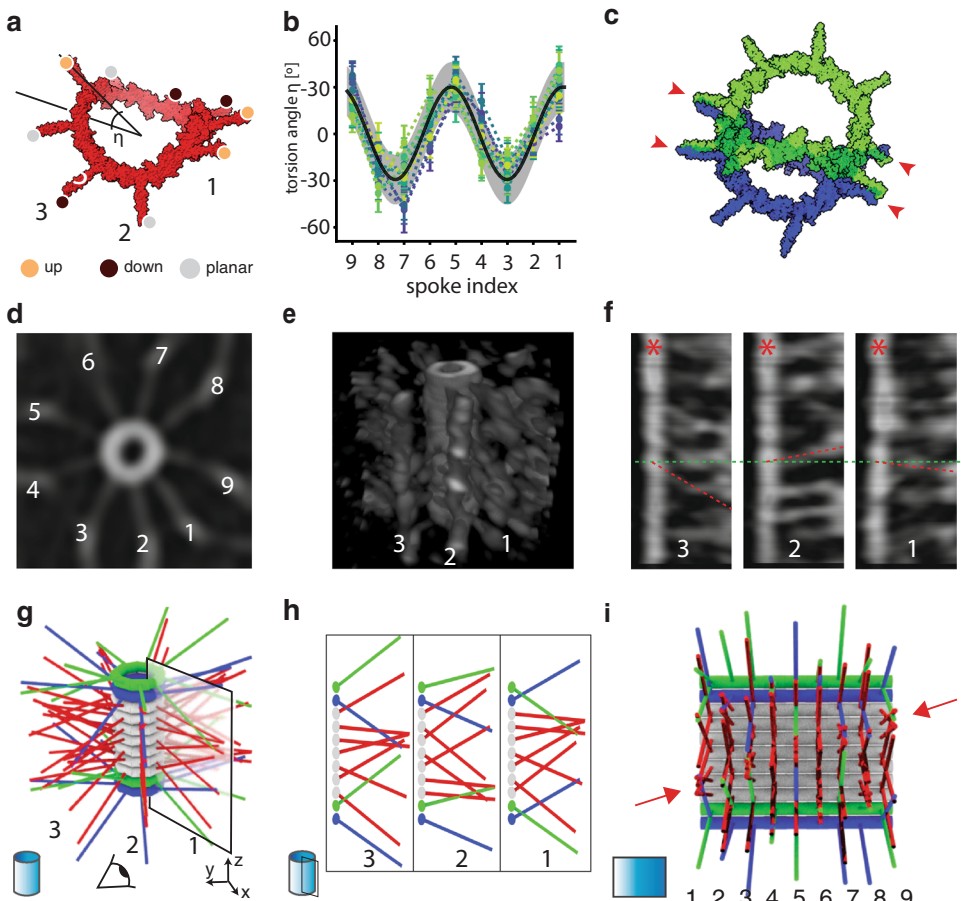

**Fig. 6 Consequences of asymmetric spoke orientation on stacking mechanism. a, b** Coarse-grained MD simulation of a virtual 9-fold SAS-6 ring enforced to close in solution (**a**). Torsion angle η of individual spokes along the ring circumference in MD simulations ($N = 10$ independent replicates) fitted by a cosine function (**b**, black line). Error bars represent the SEM for each angle in individual simulations. **c** Two successive 9-fold SAS-6 rings arranged so that spoke oscillations derived from the MD simulation are in anti-phase (**c**). Unlike in the current paired model of SAS-6 ring stacking, only the spokes marked by red arrowheads are bridged between the two successive rings in this case, leaving the others available for further interactions. **d–f** Cryo-electron tomogram of in vitro assembled SAS-6 stack (from[39]) without symmetrization or averaging. Individual sectors of a stack are identified in the top view maximum projection (**d**). Corresponding 3D volumetric representation of the stack with numbers indicating the three best resolved sectors in the front (**e**). Projected side views of these three sectors do not exhibit the regular arrangement expected from the paired model, as evidenced by spokes (dashed red lines) at the same height (dashed green line) not having the same orientation (**f**); the central hub is marked by red asterisks. **g–i** Geometrical model of multiple ring stacking, adjusting the relative in plane angle to maximize spoke contacts, leads to a regular but non-symmetrical spoke 3D arrangement (**g**). The regular repetition of angles along the ring circumference leads to a position-dependent distribution of spoke angles η in each sector, best visible in the side view (**h**). Lateral visualization unwrapped to obtain a complete view of the cartwheel surface, showing a chiral arrangement of spokes (diagonal spoke merging indicated by red arrows) (**i**).

the remarkable architecture of the organelle was unveiled by electron microscopy in the late 1950's. How do molecular interactions that occur at the nanometer length scale translate into the stereotyped architecture of the nascent centriole at the organelle scale, including its orthogonal emergence from a surface surrounding the resident centrioles and its characteristic chirality? Our work provides initial elements of answer to these questions. By analyzing the self-assembly of the SAS-6 ring polymer at the single molecule level, we found that ring assembly is drastically favored by the surface, where the polymerization reaction follows a coagulation-fragmentation process. The surface also ensures that SAS-6 homodimers assemble predominantly into 9-fold rings. Moreover, SAS-6 rings are not perfectly flat on the surface and instead exhibit non-random torsional angles between homodimers. This prompted us to revisit the current model of SAS-6 ring stacking and to propose an alternative mechanism whereby the breaking of radial symmetry inherent to SAS-6 ring

polymer formation could contribute to the chirality of the centriole organelle.

Our findings demonstrate that surface-guided SAS-6 self-assembly is a coagulation-fragmentation system in which all intermediates react with one another, with dynamic ring opening and closing events being critical for robust generation of 9-fold symmetrical rings. What could be the advantage of such a flexible system compared to one where only individual homodimers could be added or subtracted? Interestingly, coagulation-fragmentation reactions converge towards equilibrium much faster than individual incremental mechanisms, thereby accelerating polymerization kinetics by up to several orders of magnitude[40]. Such acceleration is anticipated to be particularly important for centriole assembly in systems such as early embryos, where organelle biogenesis can occur within minutes[41]. Faster polymerization might also be important to decouple the time scale of ring polymer assembly from that of ring stacking[42],

so that ring formation is completed before stacking starts. However, such a coagulation-fragmentation system could result in higher centriole number following increased levels of SAS-6 on the surface, which may explain why HsSAS-6 overexpression leads to extra centrioles in tissue culture cells[15], and perhaps also in disease conditions. Furthermore, the behavior of coagulation-fragmentation systems can be modulated in a substantial manner by the addition of interacting components[40]. In the case of centriole biogenesis, it will be interesting for instance to investigate the impact of molecular partners such as Cep135/Bld10p and STIL on the SAS-6 coagulation-fragmentation system.

Our findings establish also that ring closing is a reversible process, and not an endpoint of the SAS-6 self-assembly reaction as could have been anticipated. This is in line with experiments in *Drosophila* where concomitant overexpression expression of wild-type DmSAS-6 and a non-oligomerizing variant of the protein does not prevent cartwheel formation, potentially reflecting reversibility of unproductive interactions[43]. In our work, we demonstrate that reversibility is key for favoring assembly of 9-fold ring polymers. In general, reversibility in macromolecular assembly reactions minimizes the likelihood of kinetic traps[44], as exemplified during virus capsid assembly[45,46]. We show here that the average symmetry of SAS-6 ring polymers is robust to changes in both protein concentration and $K_d$, demonstrating that reversible ring closing and opening is critical to avoid kinetic trapping also in the case of centriole assembly. This raises the intriguing possibility that weak and reversible interaction between SAS-6 homodimers has been favored during evolution to minimize trapping of ring polymers in incorrect symmetries. In essence, this can be viewed as a proofreading mechanism embedded in the coagulation-fragmentation system that favors the output of assemblies with proper symmetry.

Our work reveals that the presence of a surface makes two fundamental contributions at the onset of centriole assembly. First, the surface catalyzes SAS-6 self-assembly, shifting the equilibrium by a factor $\sim 10^4$ from that in solution. This equilibrium shift provides a potential rationale for why SAS-6 ring polymers form exclusively on the torus of the resident centriole and not in the cytoplasm. Intriguingly, FRAP experiments in emerging centrioles of *Drosophila* embryos indicate that SAS-6 is incorporated into the cartwheel on the Asl torus, from the proximal end of the nascent organelle[47], potentially reflecting the catalytic impact of such a surface. More generally, the equilibrium shift unveiled here reinforces the importance of the interplay between cytoplasmic and surface compartments in cellular diffusion-reaction systems (reviewed in[48]), as seen for instance with the catalytic effect of membranes in promoting assembly of macromolecular complexes such as pore forming toxins[49,50]. Our findings establish that these physico-chemical principles apply not only for interactions between cytosol and membranes, but also at the space scale of organelle assembly. In the cellular context, the torus surface is expected to efficiently restrict SAS-6 ring polymer assembly to that location. Given that this surface is parallel to the long axis of the resident centriole (see Fig. 1a), such surface-mediated catalysis could readily explain how the nascent centriole systematically emerges in an orthogonal fashion.

The second fundamental contribution of the surface revealed by our work is breaking symmetry of the SAS-6 ring polymer in the vertical direction. The crystal structure of SAS-6 indicates that the polymer has an inherent helical propensity, a conclusion reinforced by MD simulations and the observation of S-shaped configurations in the PORT-HS-AFM data set. Suggestively, such a helical arrangement is reminiscent of that adopted by the structurally related proteins CCDC61 and XRCC4[51], as well as of the steep SAS-6 spiral proposed in the divergent *C. elegans* centriole[52]. Our findings provide a means to reconcile features of these evolutionary related proteins by considering that the presence of a surface changes the preferred topology of the polymer from a helix to a ring. A geometrical consequence of an inherently helical propensity is that surface-constrained SAS-6 ring polymers exhibit spoke asymmetry, which we find are present in cartwheel stacks generated in vitro. This led us to revise the current paired model for SAS-6 stacking, proposing instead that spokes from a given ring are interleaved with those from the ring above and below. The large number of possible interleaving geometries might explain the apparent inter-species variability in cartwheel stacking arrangements[37,38]. Regardless, such an interleaved stacking mechanism is expected to confer superior mechanical stability to the cartwheel. We note also that having regularly oscillating spoke angles within each ring breaks the mirror symmetry of SAS-6 stacks and thus results in a chiral cartwheel element. It is tempting to speculate that this might ultimately lead to the signature chirality of the microtubules in the centriole organelle and the cilia they template.

## Methods

**Protein expression and purification**. His-tagged CrSAS-6[NL] spanning amino acids 1–503 of the protein (see Supplementary Fig. 1a)[9] was expressed in the *Escherichia coli* strain BL21(DE3) (Stratagene). Bacteria were grown at 37 °C in lysogeny broth (LB) supplemented with kanamycin (50 μg/mL) up to an absorption $A_{600}$ of 0.7, when 0.3 mM IPTG (isopropyl β-D-1-thiogalactopyranoside) was added to induce protein expression overnight at 18 °C with shaking at 210 rpm. Cells were collected by centrifuging at 4,000 *g* for 20 min at 4 °C (JLA-9.1000, Beckman Coulter). The bacterial pellet was resuspended in 50 mM Tris-HCl pH 8.0, 400 mM NaCl, 20 mM Imidazole pH 8.0, 3 mM β-mercaptoethanol, 50 Units DNAseI, 3 mM MgCl₂, protease inhibitors (Complete mini EDTA-free, Roche) and 1% v/v Tween 20, and lysed by sonication on ice (100 cycles of 3 s pulses followed by 12 s breaks). Cellular debris were removed by centrifugation at 49,000*g* (JA 25.50., Beckman Coulter) for 1 h at 4 °C. For nickel purification, the cleared lysate was loaded at 4 °C on an HisTrap HP Ni²⁺–Sepharose column (GE Healthcare) following the manufacturer's instructions and further purified by size exclusion chromatography (SEC) using a HiLoad Superdex 200 16/60 column (GE Healthcare) equilibrated in 20 mM Tris-HCl pH 7.4, 150 mM KCl. PreScission protease was added to remove the tag, followed by an additional reverse HisTrap step to remove un-cleaved protein prior to SEC. After verifying purity with Coomassie stained SDS-PAGE (see Supplementary Fig. 1b), the sample was concentrated to 1–3 mg/mL, depending on the specific preparation, with an Amicon ultra filter unit with a 10 kDa cut-off, snap-frozen in liquid nitrogen as aliquots and stored at −20 °C.

**Modeling of SAS-6[NL]**. The model of CrSAS-6[NL] shown in Supplementary Fig. 1 was generated by combining the crystal structure of CrSAS-6[6HR] with a coiled-coil generated by CCBuider 2.0[53]. The coiled-coil region in the CrSAS-6[6HR] structure was analyzed in TWISTER[54] to extract the parameters (radius = 6.7 Å, pitch = 156.3 Å, interface angle = 21.13°) used in CCBuilder 2.0.

**CD Spectroscopy of SAS-6[NL]**. CD spectra in Supplementary Fig. 1d were collected at 20 °C using a Chirascan spectropolarimeter (AppliedPhotophysics) with a 0.1 cm path length cuvette at varying protein concentrations in 20 mM Pottasium Phosphate pH 8.0, 150 mM NaF. The dissociation of CrSAS-6[NL] was determined by monitoring the CD signal at 222 nm after buffer signal subtraction, and by fitting the concentration-dependent mean residue elipticity at 222 nm with a two-state association model.

**PORT-HS-AFM imaging**. PORT-HS-AFM imaging was performed on a custom-built high-speed setup with a head scanner and controller assembled on a spring-based vibration isolator (BM4, Minus-K), around a commercial Multimode 8 AFM base, combined with a piezo amplifier (Techproject), as previously described[27]. The entire setup was kept at 6–10 °C by placing the microscope base, scanner and head into a low-vibration cooler (WL450F-220-FL, Swisscave). Cantilever tips were electron-beam deposited from a carbon precursor (Tetradecane, Sigma-Aldrich 172456) onto the existing silicon nitride tips (BL-AC10DS, Olympus) to reduce protein adhesion to the cantilever and allow re-use. For imaging, 60 μl of filtered imaging buffer (20 mM Tris-HCl, 150 mM KCl, pH 7.4) was injected into the cantilever holder of the head. After aligning the lasers, the head was placed onto the scanner directly after cleaving the glued-on 3 mm Muscovite mica disk (Electron Microscopy Sciences, Hatfield). Purified CrSAS-6[NL] was diluted in imaging buffer to 60 μg/mL (~10.5 μM). After pre-loading the Hamilton syringe with 15 μl imaging buffer to compensate for the dead volume, 5 μl of the diluted sample was directly injected in the liquid cell already containing 60 μL of buffer, reaching a final concentration of 4 μg/mL (~120 nM). For measurement of ring opening and

closing lifetimes, a 10 µL drop of purified CrSAS-6[NL] diluted to 0.48 µg/mL (~14 nM) was deposited on freshly cleaved mica covered to avoid evaporation, incubated for 40 min at 4–6 °C, and then rinsed three times by gently exchanging the buffer through pipetting in imaging buffer, before imaging using the AFM head. Scanning was performed at 100 Hz (100kHz PORT rate) covering 512 pixels × 256 lines (for an 800 nm total scan size), corresponding to 2.56 s frame$^{-1}$. Background correction was performed after each frame, as previously described[27].

PORT-HS-AFM movies were processed with a custom written plugin in Gwyddion[55]: each image was subjected to plane leveling, initial match line (median of differences row alignment), and background subtraction (flatten base). Images were further subjected to a conservative denoise filter (2 pixels) and saved in the tiff file format.

**Opening and closing lifetime analysis.** To automatically detect the lifetimes of open and closed rings, 242 PORT-HS-AFM individual ring sequences were first manually selected from movies exhibiting opening and closing events (see Supplementary Fig. 2a). The movies were imported in IgorPro (WaveMetrics) for further analysis. For each ring, a circular fitting was performed at each frame by computing the sum over a circular profile centered in $(X_C, Y_C)$ and of radius R for all $X_c$ and $Y_c$, and varying from the center of the image ±20 pixels in $(X_c, Y_C)$ to allow for residual imaging drift and ring mobility, as well as from $R_{min} = 5$ and $R_{max} = 10$ pixels with a step size of 1 pixel. The set of $(X_c, Y_C, R)$ values maximizing the sum of values above a threshold determined empirically by visual inspection of the final fit and corresponding to 2.7 nm in height were selected along the circle (see Supplementary Fig. 2b, red ring). The corresponding profiles were plotted over time, and frames for which each point was above the minimal threshold were classified as closed (see Supplementary Fig. 2c). For each automatically detected opening and closing event, a manual validation step was performed thereafter and ring symmetry determined from the number of emanating spokes, as well as from the ring diameter. Two types of frame sequences were retained for computing lifetimes: (i) initially closed rings that opened and then closed again at a later time, without change in symmetry (lifetime open); (ii) initially opened rings that closed and then opened again at a later time, without change in symmetry (lifetime closed).

**Oligomer segmentation and classification.** Fiji[56] was utilized to combine images from PORT-HS-AFM movies in a single tiff stack, which was then loaded in the machine-learning software Ilastik[31]. In Ilastik, first, brush strokes were drawn to separate each image into 3 channels based on the following features: (1) background; (2) heads; (3) spokes. The corresponding pixel predictions maps were then utilized for the final segmentation, whereby a simple threshold ($T = 0.5$) in the head channels was applied to segment the different oligomeric entities. Only the finally segmented head domains where utilized by Ilastik for the classification, the spokes being solely a guide for the user in providing the training set. This classification process was then utilized to manually label objects and train a classifier using all features for the following 14 classes; oligomers containing 1–10 homodimers, as well as closed rings with 7–10 homodimers. The resulting classification, consisting of the probability of each object to be assigned to each class, was exported as a hd5 file for further processing in IgorPro.

**Kinetics fitting procedure.** The output of the classification was used to compute the number of assemblies of each class in every frame as $N_j(t) = \sum_i P_j^i(t)$, i.e. the probability $P_j^i(t)$ that an object i is in a class j, with i spanning all detected objects in that frame. The error in classification was estimated as $E_j(t) = \sqrt{N_j(t) * \frac{N(t) - N_j(t)}{N(t)}}$, which was derived under the assumption that the variance of a single bin in a multinomial distribution under normal approximation is $Var(N_j(t)) = N(t) * P_j(t)(1 - P_j(t))$. The resulting kinetics for each PORT-HS-AFM movie were also used to compute the overall concentration of homodimers on the surface as the weighted sum of all classes, i.e. $N_{tot}(t) = \sum_{j=1}^{10} j * N_j(t) + \sum_{j=11}^{14}(j - 4) * N_j(t)$. We additionally computed the average polymerization degree as $\bar{N}(t) = N_{tot}(t)/N(t)$ (see Supplementary Fig. 3).

Usually coagulation-fragmentation equations are considered for closed systems in which the overall mass of the polymers is conserved (see Supplementary Note 1 for a more detailed discussion). However, here there is an open system with a constant increase in polymer concentration over time due to homodimer influx into the chamber and subsequent adsorption to the mica surface; such an increase must be taken into account for the fit. In principle, this increase could have been determined experimentally in each frame. However, this would have unavoidablly introduced noise associated with experimental extrapolations, which would have resulted in non-smooth boundary conditions in the numerical solution of the ordinary differential equations. Fot this reason, in the fitting procedure, we opted for including the effect of the increase in total concentration over time with an additional term in the $C_1$ concentration (i.e. the concentration of SAS-6 homodimers), accounting for diffusion in solution after injection, as well as for adsorption and desorption from the surface.

For the concentration in solution, we approximated the temporal dependence as:

$$C_{sol} = C * (Erf(S(t - t_0)) + 1)$$

The error function recapitulates the fast increase and subsequent saturation very well (see Supplementary Fig. 3). For the surface processes we assume:

$$\frac{dC_{Surf}}{dt} = A * C_{sol} * (T - C_{surf}) - D * C_1$$

Thus the increase is driven by the concentration in solution $C_{sol}$ and limited by the capacity T of the surface. The surface-adsorbed homodimers then feed into the coagulation-fragmentation equations through $C_1$. These equations were then fitted for each PORT-HS-AFM movie to derive the parameters C, S, A, T and D. These parameters were then fixed for the ensuing fitting using the global fit package of IgorPro by fitting all time dependent curves for the 14 species with the same $k_{on}$ and $k_{off}$. It should be noted that the equation for adsorption/desorption does not necessarily reflect the actual adsorption/desorption process and is used here solely as an effective equation fitting to derive accurately the trend of the surface concentration over time.

**Model comparison.** To test if the proposed coagulation-fragmentation model is adequate to represent the SAS-6 ring assembly reaction determined experimentally, we fitted both a simpler model and a more complex one to the measured trajectories. The simplest model that can be considered is one in which only individual homodimers can be added or removed. In this case, also known as the Becker-Döring model[57], the equations simplify to:

$$\frac{dC_1}{dt} = -2k_{on}C_1 \sum_{l=1}^{9} C_l + 2 \cdot k_{off} \cdot \sum_{l=2}^{10} C_l$$

$$\frac{dC_j}{dt} = 2k_{on}C_1 C_{j-1} - 2k_{on}C_1 C_j + 2k_{off}C_{j+1} - 2k_{off}C_j$$
$$- k_{close}^j \cdot \sum_{l=7}^{10}(\delta_{j,l} \cdot \bar{C}_j) + k_{open}^j \cdot \sum_{l=7}^{10}(\delta_{j,l} \cdot \bar{C}_j)$$

The corresponding fit has a higher reduced Chi-square than our model ($\chi_\nu^2 = 3.37$ versus 1.01), indicating poorer performance. This was expected as visual inspection clearly showed coagulation and fragmentation event of differently sized oligomers. The most complex model would have required 58 free parameters (25 for all the possible coagulations, 25 for all the possible fragmentations and 8 for ring opening and closing at different symmetries) to allow for all possible interaction constants between all oligomeric species, rendering it computationally unfeasible. Therefore, we opted instead for an intermediate complex model for comparison. Considering that homodimers seem to be the species with the largest rotational freedom, we generated a model with intermediate complexity in which all $k_{on}$ and $k_{off}$ are identical except those involving encounters with homodimers, which are treated with $k_{on}^{1j}c_j$, adding 9 more free parameters compared to the simple model. As expected, the resulting fit yielded a slightly lower reduced Chi-square ($\chi_\nu^2 = 0.99$). However, the additional 9 free parameters fitted on a single movie did not increase the predictive power of the model for the other movies, since the Chi-square between experimental data and the predicted dynamics was not significantly differently ($\hat{\chi}_{comp}^2 = 12 \pm 5$ versus $\hat{\chi}_{simple}^2 = 14 \pm 6$), thus validating the choice of the simpler model.

**Fitting of angle between homodimers.** To find the best fitting chain of open SAS-6 oligomeric species, we adopted a three-step approach. First, images of oligomeric species were extracted from the Ilastik output, together with their class prediction. For each particle, a $50 \times 50$ pixels$^2$ region centered around the geometrical center of the oligomer was cropped and the maximum height normalized. This region was subsequently masked using the head segmentation described in the Oligomer segmentation and classification section, so that only the head domain was retained. In a second step, a custom algorithm written in Python was applied to assign the initial chain position and angles for each cropped oligomer. In brief, the algorithm consists of an optimization performed over the angles between the segments of a continuous line to find the angles that best match the observed image. Initially a chain of N connected vertices was generated at an angle of 40°, where N is the oligomeric state derived from Ilastik. An image was then generated with a line connecting these vertices; this line was then blurred with a Gaussian filter to mimic the broadening in the resolution due to imaging. This artificially generated image was then aligned to the original one with the template matching algorithm matchtemplate (OpenCV library), and thus the best relative XY position and corresponding image cross-correlation value identified. The initial angles were then modified by a vector $\delta\bar{\alpha}$ consisting of all inter-segment angles plus the initial angle, thus defining the overall rotation of the entire line. The vector $\delta\bar{\alpha}_{opt}$ that maximizes the correlation then served as output of the optimization algorithm. The operation was repeated for N−1 and N+1 connected vertices to verify that the maximal correlation corresponded to the predicted class. If that was not the case, the process was iteratively repeated, starting from the N* maximizing the correlation. Given the stochastic nature of minimum finding algorithms in general, the line fitting procedure was further refined in a third step by an exhaustive search algorithm

written in IgorPro, starting from the solution found in the previous step, which was subjected to iterative small angle modifications spanning all angles ±10° (every 1°). At each iteration with an angle modification, the two successive angles were re-calculated to solve the corresponding four-bar linkage problem[58]. The angle over which the image profile was maximized was retained and kept as the starting condition for the following angle, until angles were stable.

**Spoke mobility analysis from PORT-HS-AFM.** To analyze spoke mobility in PORT-HS-AFM experiments, we first manually selected 10 9-fold rings that remained closed for >20 frames. For each frame of each ring, we then generated a circular profile, fitting the head domains with the algorithm described in the Opening and closing lifetime analysis section. From those positions, we generated a second profile with a 10 nm larger radius (see Supplementary Fig. 5j). These outer profiles were then used to generate kymographs that were manually divided into 9 regions corresponding to the 9 spokes (see Supplementary Fig. 5k). The location of each spoke was then tracked across the kymograph by finding the maximum height within the relevant region in each frame. A time window of 13 frames was then picked randomly from each kymograph to avoid averaging spoke behavior that might have changed over longer times. Within this time window, discrete spoke positions in each frame were used to compute the average displacement over time from the average spoke position. The histogram of the average square displacement for all spokes in the 10 selected rings for three random choice of time window positioning is shown in Supplementary Fig. 5l. To verify the stability of this analysis, we computed at varying threshold the number of spokes classified as immobile, as well as the number of equally oriented spokes (see Supplementary Fig. 5m, green lines). We then confronted this computation with the number of equally oriented spokes calculated from a random distribution of immobile and mobile spokes, using varying probabilities for each spoke to be mobile. We found that the experimentally measured distribution differed significantly from a random one over a sizeable range (see Supplementary Fig. 5n).

**Molecular dynamics.** We modeled a solution structure of the *Chlamydomonas* SAS-6[6HR] nonamer based on the two PDB structures 3Q0Y -containing head domains connected by their N-terminal interface, and 3Q0X -containing a homodimer including 6 heptad repeats of the coiled-coil interface[9]. Since 3Q0Y has a higher resolution (2.1 Å compared to 3.02 Å of 3Q0X), we built a homodimer homology model with improved resolution from the 3Q0Y head domain and the 3Q0X tail domain by aligning the two structures using UCSF Chimera (version 1.12[59]) and Swiss Model in user template mode[60]. To generate a nonameric assembly, we then aligned a head domain of 3Q0Y to a head domain of the new hybrid homodimer model and used this as an N-terminal interface template to align a second homodimer structure. This procedure was repeated until a complete nonamer was generated, which exhibited a spiral structure. The martinize.py script (version 2.6 [61]) was used to obtain the structure in coarse-grain representation.

To build a flat nonameric SAS-6 ring, we used Gromacs tools version 2018 [62,63] to align one coarse-grained homodimer with the xy-plane. We copied this homodimer and rotated each subsequent copy by 40° within this plane until 9 homodimers were arranged in a flat circle, with the N-terminal head interfaces of two neighboring homodimers close to each other in the center and the 6 heptad repeats of the coiled-coil pointing outwards. We then utilized an elastic network to close the ring. Elastic networks are commonly utilized in coarse-grain simulations to compensate for the loss of detailed non-covalent interactions[64]. We obtained the elastic network by coarse-graining a SAS-6[6HR] dimer using martinize.py, with 0.5 nm and 0.9 nm for the lower and upper cut-offs, respectively, and replicating it to have the same network on all nine interfaces. The same elastic network was also imposed on the spiral structure for stabilization. To pull the 9 homodimers arranged in the above manner into a closed ring, we ran a 1 ns simulation with 5% (25 kJ mol⁻¹ nm⁻²) of the original force constant (500 kJ mol⁻¹ nm⁻²) on the elastic network. The complete simulation conditions are listed below. To get SAS-6[6HR] oligomers smaller than a nonamer from both the flat ring and the spiral shaped structure, the required number of homodimers was deleted from the respective nonameric structures to obtain the desired size. For simulations including a generic surface, we only used the structures derived from the flat ring; the surface was generated using VMD (version 1.9.3[65]) Nanotube Builder with carbon beads placed with graphene geometry at a distance of 0.282 nm.

All coarse-grain simulations were performed with Gromacs version 2018.5 along with the Martini force field (version 2.2)[61–63]. We based the surface parametrization on non-bonded C1 beads, but decreased the interaction strength between protein and surface beads by two levels in the Martini interactions table, and increased the water-surface interactions from a repulsive to an intermediate interactions level. This parametrization has a lower hydrophobicity, and allows diffusion on the time scale of the MD simulations, resulting in a surface with medium-level adhesion strength of the protein, i.e. a generic flat template for SAS-6[6HR]. For simulations of SAS-6[6HR] in solution, we placed the protein structures in a cubic box with 3 nm minimal distance to the nearest box edge. For simulations on a surface, we chose the x- and y-dimensions to fit the generic surface (46 × 46 nm). The size in the z-direction was 20 nm. All systems were solvated with Martini water, spiked with 10% anti-freeze water and neutralized by the addition of 150 mM NaCl. We energy minimized the systems using the steepest descent algorithm with a step size of 0.01 nm until energy convergence and a

maximal force below 100 kJ mol⁻¹ nm⁻¹. Surface beads were frozen during minimization and position restrained by a harmonic potential with a 1000 kJ mol⁻¹ nm⁻² force constant during all other equilibration and production simulation steps.

We equilibrated the minimized structures for 1 ns in the NVT ensemble, followed by 200 ps in the NPT ensemble with a 2 fs integration time step, as well as 1 ns in the NPT ensemble with a 10 fs time step and a 1000 kJ mol⁻¹ nm⁻² force constant on protein position restraints. To allow further relaxation of the structures, we ran two 30 ns simulations with a 30 fs time step and decreased force constants for position restraints on protein beads of 200 and 100 kJ mol⁻¹ nm⁻², respectively. The production simulations each continued for 1.5 μs with a 30 fs integration time step without position restraints on protein beads. Periodic boundary conditions were applied in all simulations. We kept the temperature at 320 K using the velocity-rescaling thermostat with a coupling constant of 1.0 ps. Isotropic pressure coupling was ensured by the Parrinello-Rahman barostat with a compressibility of $3.0 \times 10^{-4}$ bar⁻¹ and a coupling constant of 12.0 ps. We treated long-range interactions using a reaction-field, assuming a dielectric constant of infinity beyond a cutoff of 1.1 nm and a relative dielectric constant of 15. Van-der-Waals interactions were cut off after 1.1 nm. The Verlet neighbor list was updated every 20 integration steps, and we set the Verlet-buffer-tolerance to 0.005 kJ mol⁻¹ ps⁻¹.

We prepared the SAS-6[6HR] dimer structure in all-atom resolution based on the crystal structures as described above, and placed it in a dodecahedron simulation box with a minimal distance from the nearest protein atom to the box edge of 2 nm. We solvated with TIP3P water and neutralized by the addition of 150 mM NaCl. We energy minimized the system using the steepest descent minimization with a step size of 0.01 nm until energy convergence and a maximal force below 1000 kJ mol⁻¹ nm⁻¹. For all simulations at all-atom resolution, the Amber99sb*ildnp force field[66] was used with a time step of 2 fs for the leap frog integrator. We equilibrated for 100 ps in the NVT ensemble followed by 1ns in the NPT ensemble. A constant temperature of 300 K and pressure of 1 bar was ensured respectively by the velocity-rescaling thermostat with a coupling constant of 0.1 ps and Berendsen barostat with a coupling constant of 2 ps and a compressibility of $4.5 \times 10^{-5}$ bar⁻¹. We cut off all short-range interactions after a distance of 1 nm and treated the long-range electrostatic interactions using the Particle-Mesh-Ewald method. The Verlet neighbor list was updated every 20 integration steps and periodic boundary conditions were in place. All bonds involving hydrogen atoms were restrained using the LINCS algorithm. Positions restraints with a 1000 kJ mol⁻¹ nm⁻² force constant, which we used to restrain all protein heavy atoms during equilibration, were released for the subsequent production simulation runs. These continued for 200 ns each with the same conditions as applied to the equilibration steps except that we used the Parrinello-Rahman barostat with a coupling constant of 5 ps for isotropic pressure coupling.

For the analysis of angles between neighboring homodimers, vector $\bar{x}$ describes the orientation of the head domain and is defined as the vector connecting the center of mass of both head domains, while vector $\bar{c}$ describes the orientation of the 6 heptad repeats and connects residue F153, which marks the beginning of the coiled-coil between the two head domains of a homodimer, and residue Q391 close to the C-terminal end of the 6 heptad repeats (Supplementary Fig. 4). Vector positions were extracted using Gromacs (version 2018) tools and subsequent geometric analysis was done using custom python 3 scripts. The in-plane angles describe hinge movement of two homodimers a and b relative to each other. The in-plane angle $\alpha$ is defined as the angle between $\bar{x}_a$ and $\bar{x}_{b'}$, with $\bar{x}_{b'}$ being the projection of $\bar{x}_b$ into the plane defined by $\bar{x}_a$ and $\bar{c}_a$. The torsion angle $\eta$ is defined as the angle between $\bar{c}_a$ and the surface.

To analyze whether the distribution of spokes is random, we iterated over the neighbors of each spoke and calculated the probability for the neighbors to be in the same orientation, i.e. bound or not bound, based on the overall percentage of bound or unbound spokes. We compared the resulting number of spokes expected to have the same orientation as their neighbor given a random distribution to the number of spokes that were actually observed to have the same orientation.

Spoke mobility was calculated by a root mean square fluctuation (RMSF):

$$RMSF = \sqrt{\langle (r_i - \langle r_i \rangle)^2 \rangle}$$

with $r_i$ being the position of residue Q391 within the xy-plane or along the z-axis. The spokes were classified as bound or not bound based on a position cut-off in z-direction.

**Tomogram processing.** Individual 9-fold cartwheels (N = 6) were manually isolated from published reconstructed tomograms[39]. The tomogram was then processed in FiJi[56]. The processing consisted in successive re-slicing (as per the Stack, Reslice operation in FiJi) and rotation (without interpolation) to manually align the cartwheel axis with the stack z-axis. The intensity of the entire stack was filtered (with the bandpass filter operation, filtering structure down to 200 px up to 20 px) and subsequently inverted (Image, Invert operation) and background subtracted (with a rolling bar of 20 px). The result of this operation is shown in Fig. 6e. Individual lines manually fitting the cartwheel sectors were then drawn on the maximum intensity projection (Fig. 6d) and used to create corresponding re-sliced selections along the z-direction (Fig. 6e).

**Geometrical stacking modeling**. The stacking model was simulated with custom written functions in IgorPro (WaveMetrics). Each ring spoke angle was assigned to a matrix $M_{i,j}$ of dimension $9 \times 10$ (9 spoke per ring and 10 rings) according to the fit of the MD simulations data (see Fig. 6): $\eta_i = a_0 + a_1 \cos(\omega * i + \phi)$, with $a_0 = 0.41, a_1 = 29.74, \phi = -0.73$. The matrix was filled starting from $M_{i,0} = \eta_i$. Thereafter, each successive column $M_{i,j}$ was filled by generating all possible rotations and complete inversions of $\eta_i$ (for a total of 18 combinations), and of these the $\bar{\eta}_i$ that maximized the number of spoke crossing with the $M_{i,j-1}$ layer was selected to populate $M_{i,j} = \bar{\eta}_i$. If more than one vector maximized the number of spoke crossing, the vector minimizing the sum of the absolute distances between the endpoint of the non-crossing spokes with the same radial position was selected.

Geometrical positions for the rendering of the models were then generated by roto-translating a two-line representation of SAS-6 dimers (one line for the head domain and one for the spoke). The roto-translation consisted of a rotation around the ring center of i*40° in plane, and of $M_{i,j}$ in the tilt axis, followed by a translation of j*(4 nm) in the direction orthogonal to the ring plane.

**Reporting summary**. Further information on research design is available in the Nature Research Reporting Summary linked to this article.

## Data availability
The data supporting the findings of this study are available from the authors on reasonable request. The MD results are deposited and publicly available at the open-access research data repository of Heidelberg University: (heiDATA(uni-heidelberg.de), permanent link: https://doi.org/10.11588/data/3NKHAY).

## Code availability
All the custom functions written for the analysis of the data contained in the manuscript are deposited and publicly available at the repository: https://github.com/UPGON/HS_AFM_SAS_6_Analysis.git (permanent link https://zenodo.org/badge/latestdoi/401994877).

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

## Acknowledgements

We thank A. Bezler, G. Garcia-Rodriguez, P. Guichard, E.A. Lemke and F. Schneider for critical reading of the manuscript. This work was supported by the European Research Council (ERC), through AdG 835322 (CENGIN) to P.G., as well as StG 307338 (NaMic) and CoG 773091 (InCell) to G.E.F. Further support came from the Swiss National Science Foundation through the European Union's Seventh Framework Programme FP7/2007-2011 under grant 200021_182562 agreement 286146 to G.E.F. N.B. was supported by the EPFL Fellows postdoctoral fellowship program funded by the European Union's Horizon 2020 Framework Program for Research and Innovation (Grant agreement 665667, MSCA-COFUND). F.G., S.d.B. and U.S.S. acknowledge funding through the Deutsche Forschungsgemeinschaft (DFG, German Research Foundation) under Germany's Excellence Strategy — 2082/1 — 390761711. Moreover, F.G. and S.d.B. acknowledge funding by the Klaus Tschira Foundation, the state of Baden-Württemberg through bwHPC, as well as the DFG through grant INST 35/1134-1 FUGG. S.d.B. thanks the Carl Zeiss Foundation for financial support.

## Author contributions

Design of the work: N.B., P.G., G.F., F.G. and U.S. Data acquisition: N.B., A.N., S.d.B., G.N.H., T.H. and F.S. Instrument development: A.N., C.B. and S.A. Data analysis: N.B., A.N. and S.v.B. Data interpretation: N.B., A.N., P.G., G.N.H., G.F., F.G., S.d.B. and U.S. Manuscript writing: N.B., P.G., G.F., F.G. and U.S.

## Competing interests

The authors declare no competing interests.
