## [Peer Review File · Nature Communications]

Kinetic and structural roles for the surface in guiding SAS-6 self-assembly to direct centriole architecture- Reviewers' Comments:

Reviewer #1:

Remarks to the Author:

Being asked by the editor to focus on the validity of the simulations, I hereby restrict my comments on this aspect of the manuscript.

In general, the simulations seem to be performed well, and clearly support the experimental data. Besides, the simulations provide key insights into the proposed noval stacking mechanism of the SAS-6 rings.

I have one major point of criticism, and a few minor ones.

The weakest part of the simulation setup is the lack of validation of the chosen approach to represent the surface. The coarse grain approach, based on the Martini model, requires finetuning of the interactions between the protein/solvent beads on the one hand, and the surface beads on the other. This is only described as a rather ad-hoc procedure, with adjustments of interaction strengths to allow diffusion of the proteins on the surface.

However, one of the key findings of the simulations is the presence of different populations of spokes - either adsorbed or desorbed from the surface. In light of lack of validation of the surface parameters, the question now is, how realistic is this finding? In addition, the conformations of the proteins (e.g., the reported angles) could well depend on the nature of the surface.

I urge the authors to perform control simulations with different surface parameters to test how robust these findings are. Even better, a more thoughtful modeling of the surface is attempted (see for instance recent work by Piskorz et al., J. Phys. Chem C, 2019).

In connection to this validation, also the possible effect of the strength and extension of the elastic network on these properties should be evaluated and discussed.

Other, minor remarks:

Martini force field version 2.6 is mentioned, but this version is non-existent (they may be confused with the version numbering of the martinize script?). Besides, no reference is given to the actual Martini protein force field that was used.

It would be really helpful to include somewhere a Figure of the actual SAS-6 protein (dimer) structure.

Reviewer #2:

Remarks to the Author:

In this paper Banterle et al., use highspeed AFM to examine the kinetics of Sas-6 ring assembly in vitro. The paper is quite complicated, but it is generally well written, the data is very novel and of high quality, and the authors come to several important and provocative conclusions. I think the paper will be of great interest to the centriole community, and also to those interested in understanding and modelling organelle assembly more generally. I am therefore strongly supportive of publication in Nature Communications, although I have a number of points the authors should consider prior to final acceptance.

Major points:

1. Extrapolating the kinetics derived from the in vitro data described here to the situation in vivo is likely to be more complicated than the authors suggest. The authors estimate that in their in vitro experiments the surface shifts the equilibrium towards oligomerisation by a factor of $\sim 10^4$ (compared to solution) and then do a back-of-the-envelope calculation to show that if the same were true for the surface of a centriole, then the kinetics observed here would be sufficient to explain why daughter centrioles only form at the surface of the mother centriole. This calculation does not seem particularly useful to me. First, the 100 Sas-6 homodimers estimated to be present at human centrioles are probably mostly (usually entirely?) in the single stacked cartwheel of the daughter centriole, not randomly spread across the entire centriole surface. Second, in the in vitro experiments, lots of cartwheels assemble on the surface, whereas in vivo, cartwheel assembly is restricted to a single site on the surface of the centriole, where it is thought that Plk4 promotes cartwheel assembly.

This does not in any way detract from the importance of making these in vitro measurements, but I think the authors should tell readers more forcefully that it is not necessarily straightforward to infer kinetics at the single site of cartwheel assembly in vivo based on these in vitro measurements with a single purified component.

2. In line 267 and line 275 the authors phrasing makes it sound as though it is very mysterious how a molecule that is intrinsically helical might form a closed ring when assembled on a surface. I may be missing something here, but to me this seems obvious, as the interaction with the flat surface is simply strong enough to force the ring to adopt a flat conformation. The authors state that the observation of S-shaped conformations supports that idea that these closed rings would prefer to be helical, but in the movie I could see one such structure forming from the fusion of two open rings. If this can happen, does the existence of such structures still suggest that the structures prefer to be helical?

3. I found the last section of the results on how the asymmetry of Sas-6 ring polymers may guide ring-stacking a little confusing. I'm not an expert in molecular dynamic simulations so I did not fully understand the description of how this was performed nor the assumptions about the molecular interactions that are required to drive the apparent non-random orientation of the spokes in the closed rings. I assume there is some cooperativity, such that the angle of the spokes in one dimer influences the angles of the spokes in nearby dimers (although this can't be too strong or the ring would be trapped in an "all up" or "all down" orientation). But my real confusion was in trying to understand how the asymmetry of the rings formed in solution influences the assembly process to generate asymmetries in the stack—as the authors just told me that the assembly process is not driven by rings formed in solution but rather by oligomers assembling into rings on a flat surface. So, for example, the interactions shown in Fig.6C (between two undulating rings in solution) are not likely to occur on the flat surface.

I guess my confusion in part comes from not understanding how ring stacking occurs in vitro. Does a ring form on the surface and then oligomers assemble on top of this (using the tail of the spokes to guide assembly, which seems unlikely)? Or, are the authors arguing that rings can actually form in solution as well, and these can then stack on top of the surface rings (perhaps as depicted in Fig.6C). In addition, the authors state that "to examine ring-stacking in the cellular context" (line 318) they looked at in vitro assembled Sas-6 stacks. I don't think the authors can be certain that the mechanisms of stacking in vitro necessarily reflect those in vivo, so again they should be more cautious in how they describe this.

Minor Points:

1. The authors should explain in the main text that the Sas-6 protein they use here is a truncation, rather than referring to a previous publication. This is an important point that readers should be fully aware of.

2. The authors nicely show that Sas-6 self-assembly is reversible in vitro. I think Cottee et al., (eLife, 2015) inferred that this must also be the case in flies in vivo (as a mix of WT and head-domain mutant forms of Sas-6 did not dominantly kill cartwheel assembly), which might be worth mentioning as it supports the authors in vitro conclusions.

3. Supp Fig. 3. In h) the authors say Fig.3b but I think mean Fig.3c.

4. The red line and angle shown in Fig.S4d is not explained in the legend and is buried in the Methods.

Reviewer #3:

Remarks to the Author:

The authors present an elegant AFM work in which they were able to directly visualize the self-assembly of SAS-6, a key protein of the centriole organelle by the use of the high-speed (HS) time lapse AFM instrument. Importantly, they utilized the instrument termed PORT-HS-AFM capable of gentle touch of the sample. Such an instrumentation was a critical factor for reliable observation of the SAS-6 protein nanoscale dynamics, which includes association and dissociation steps along the self-assembly process. The authors acquired a dataset of SAS-6 oligomers as large as ~40, 000 species, which allowed them to quantitatively analyze the data using the oligomers assembly-disassembly model developed in the paper. Given that HS-AFM is a single molecule technique, such statistics is really impressive. Overall, the observations led the authors to a model of the self-assembly of SAS-6 oligomers with ring shape explaining their assembly in the centriole cartwheel. The experimental data are supplemented by computer modeling of the SAS-6 oligomerization, which provided additional details to the three-dimensional shape of SAS-6 rings. My few comments to the paper are listed below.

1. The time-lapse results demonstrate that SAS-6 protein have a high affinity to the mica surface, so the protein rather rapidly covers the surface. In parallel, the protein assembles in oligomers; however, their analysis of the kinetics of oligomers assembly does not include the contribution to the oligomers assembly from the protein species coming to bound oligomers from the bulk solution. Such events can be seen in the movie file. If the 2D diffusion is the predominant pathway for the oligomer's assembly, it should be justified.

2. The role of the mica surface on the shape of SAS-6 rings needs to be discussed thoroughly. Additionally, mica is not a natural environment for cells, therefore a translation of the geometrical data for the rings assembled on this surface to their shape in natural environment such as surface of torus sounds quite speculative. Two factors, the surface shape and composition can contribute to the self-assembly process and the ring shape and these factors should be discussed.

3. The asymmetry of SAS-6 rings and their assembly in cartwheel is described along with the Cryo-EM results. However, these are data published by others, so it is appropriate to move this section of the paper to the discussion section.

Reviewer #4:

Remarks to the Author:

Centriole duplication occurs proximally to a parent centriole with the daughter centriole elongating orthogonally and remaining attached to the parent until the end of cell division. SAS-6 self-assembly results in a cartwheel-shaped feature that is one of the initial steps in the

centriole duplication process. The prevalent hypothesis regarding centriole duplication places the polo-like kinase 4 (Plk4) as the key player in the initiation of the duplication process and a recruiter/regulator for the self-assembly of SAS-6. However, over-expression of SAS-6 results in supernumerary centrioles, suggesting an alternative self-assembly pathway. Banterle et al. characterized the effect of surfaces on the self-assembly of SAS-6 using HS-AFM (high-speed AFM, off-resonance mode with photothermal actuation) combined with coarse-grained molecular dynamics (CG-MD) simulations. Additionally, they provide a theoretical framework for the self-assembly process.

Authors show that, upon binding to the mica surface, SAS-6 homodimers assume spoke and head type conformation, which are capable of further self-assembly, through the head regions, into rings with protruding spokes. Furthermore, their experiments revealed that the self-assembly process is very dynamic with multiple pathways for ring assembly, ring-opening, and dissociation/fragmentation of rings and partial rings. Authors then use the obtained data to establish a theoretical framework and determine the appropriate rate constants for the self-assembly of SAS-6. Additionally, they investigated the physical reason for the observed phenomenon using CG-MD simulations.

The key takeaways are:

- Ring formation efficiency exhibits a strong dependence on effective surface concentration.
- Ring formation is reversible.
- A coagulation-fragmentation model is proposed that supposes that all intermediates react with one another, which greatly facilitates the self-assembly due to faster reaction convergence.
- Surface effects ensure that the rings are nine-fold symmetric, not flat, and exhibit non-random torsional angle between homodimers; which has implications for the stacking of SAS-6 rings in the centriole duplication process and may explain its chirality.

Overall, the manuscript expands the current knowledge about SAS-6 self-assembly with a clear narrative. In addition to the scientific merit, the methodological value presented by the semi-automatic data-processing routines (using freely available software) is also of notice and commendation. Although of high quality, the manuscript has a few minor issues:

- Authors should provide close-up movies for each type of assembly presented in Fig. 1. Similar to how the opening-closing is shown in Supp. Movie 2. Similarly, the S-shaped assembly should be presented in a separate movie.
- Supp. Fig. 3a legend: Fig. 3b should be Fig. 3c. Although, the open and closed states have been added and the panel is not exactly the same. Similar internal refs. should be checked.
- References need a thorough editing as some are incorrect (probably due to manuscript iterations). E.g. Fig. 5b, ref. 5 should be ref. 8?
- An opinion: I strongly urge the authors to publish the MD results and corresponding analysis in Scientific Data (or similar journal). Availability of an extended version of the MD data and analysis may prove helpful for other researchers with other foci.

Reviewer #5:

Remarks to the Author:

Summary

The authors investigate how SAS-6 assembles into ring polymers, which are an essential component of the centriole organelle.

The authors use high-speed atomic-force microscopy to observe the assembly process on a surface.

They use machine learning to segment and classify the various intermediate states and of the polymer assembly.

They propose a system of differential equations to describe the assembly process and go on to determine the parameters of their model based on the observed frequencies (determined via ML classification) of the different intermediate states.

Based on their observations and simulations, the authors reason that the presence of a surface is essential for the assembly of ring polymers as they occur in the centriole and go on to develop a theory for the stacking behavior of polymer rings, which is validated via cryo-electron tomography.

Scope of this Review

I really only feel qualified to comment on the machine learning aspect of the paper. I regret that I cannot comment on the other aspects.

Machine Learning Segmentation and Classification

The authors use the Ilastic software for segmentation of their AFM images, as well as for classification of the identified objects.

Ilastic is a solid choice and by now a well established tool.

Image processing of the AFM images is done in two steps.

First the images are segmented via pixel classification, that is each pixel is assigned a probability of belonging to the background, head domain, or spokes.

Based on this pixels are grouped to objects, which are then again classified by Ilastic and assigned a probability of belonging to one of the oligomer types.

I appreciate it that the authors illustrate the process in Supplementary Figure 2.

While I absolutely agree with the image processing pipeline, there is also a little room for criticism:

It looks like the classification in particular is a challenging problem with this data and far from trivial.

The authors mention in the caption for Supplementary Figure 2 that they train the classifier by manually assigning "some oligomers with clearly identifiable spoke numbers" to the corresponding class.

So it seems that at times the correct spoke number is not easily identifiable, even for a human expert.

I believe Ilastic cannot learn to identify spoke numbers in ambiguous oligomers without receiving properly annotated training data for exactly this ambiguous type oligomers.

Can the authors comment on this problem?

Are there oligomers that are not identifiable by a human?

Roughly what fraction would fall into this category?

After classification by Ilastic are these what class are these oligomers assigned to?

Do they receive a high uncertainty label, i.e. a high entropy probability distribution?

How confident are the authors that their classification is reliable?

Note that the fact that Ilastic outputs a probability distribution, i.e. reflects the uncertainty of its decision does not necessarily mean that its classifications are correct.

It is for example possible that Ilastic assigns a wrong label to an object with a high confidence.

A standard way to address such concerns about the classification quality would be perform validation on a part of the dataset that was not used during training.

That is, one could randomly choose a subset of objects that were not used during training and

manually classify them.

Then these ground truth object labels can be compared to the Ilastik prediction to compute accuracy, confusion matrix, or other metrics.

I would appreciate it if such a validation was included in the final version of the paper, or if the authors would comment on the points mentioned above.

We thank the five reviewers for their thorough analysis of our work and the useful suggestions aimed at improving the manuscript. We have addressed all the comments that have been raised through experiments, analyses and rewriting, as detailed point-by-point hereafter.

REVIEWER'S COMMENTS

Reviewer #1 (Remarks to the Author):

Being asked by the editor to focus on the validity of the simulations, I hereby restrict my comments on this aspect of the manuscript.

In general, the simulations seem to be performed well, and clearly support the experimental data. Besides, the simulations provide key insights into the proposed novel stacking mechanism of the SAS-6 rings.

I have one major point of criticism, and a few minor ones.

The weakest part of the simulation setup is the lack of validation of the chosen approach to represent the surface. The coarse grain approach, based on the Martini model, requires fine tuning of the interactions between the protein/solvent beads on the one hand, and the surface beads on the other. This is only described as a rather ad-hoc procedure, with adjustments of interaction strengths to allow diffusion of the proteins on the surface.

However, one of the key findings of the simulations is the presence of different populations of spokes - either adsorbed or desorbed from the surface. In light of lack of validation of the surface parameters, the question now is, how realistic is this finding? In addition, the conformations of the proteins (e.g., the reported angles) could well depend on the nature of the surface.

I urge the authors to perform control simulations with different surface parameters to test how robust these findings are. Even better, a more thoughtful modeling of the surface is attempted (see for instance recent work by Piskorz et al., *J. Phys. Chem C*, 2019).

We thank the reviewer for raising this important point. Prompted by her/his comment, we performed four additional sets of MD simulations, with surface parameters being varied compared to the original ones as follows: (i) + 1 or (ii) - 1 level of protein-surface interaction strength, as well as (iii) + 1 or (iv) - 1 level of surface-water interaction strength, according to the Martini interactions table (PMID: 17569554). This enabled us to explore both lower and higher protein-surface interaction strength, as well as lower and higher surface hydrophobicity. This is reported in Supplementary Figure 5 a-d and discussed in the revised manuscript (page 11). As expected, we found that the fraction of bound versus unbound spokes varies upon varying surface parameters. Importantly, in addition, all four new sets of MD simulations resulted in sizeable populations of the two kinds of spoke configurations, bound and unbound (Supplementary figure 5a, 5b). Furthermore, we found that the observed angle range is robust with regards to parameter variations (Supplementary figure 5c).

Additionally, we evaluated whether the distribution of nearest neighboring spokes being bound or unbound was random in all five MD data sets. We found that the number of equally oriented neighboring spokes was higher than that expected for a random distribution not only for the original set of parameters, but also for the two new cases where spokes were less bound to the surface (lower protein-surface interaction or higher hydrophobicity). For the two remaining configurations (higher protein-surface interaction or lower hydrophobicity), the distribution was not statistically different from random. However, the fraction of bound spokes was >60% in these two cases, and it

is of course increasingly difficult to achieve a non-random distribution as the percentage of bound spokes gets closer to 100% (the limit case being one where all spokes are bound and all nearest neighbors are identical). This additional analysis is reported in the new Supplementary Figure 5d, including its legend.

In connection to this validation, also the possible effect of the strength and extension of the elastic network on these properties should be evaluated and discussed.

As requested by the reviewer, we have also tested different elastic constants ($K_{el} = [10,50,100,250]$ $\text{kJ mol}^{-1} \text{nm}^{-2}$), in addition to the initial value of $500 \text{ kJ mol}^{-1} \text{nm}^{-2}$. This analysis is reported in the new Supplementary Figure 5 e, f. As can be seen, the RMSD values are similarly low for the 250 and 500 values, but increase for $K_{el} = [10,50,100]$ $\text{kJ mol}^{-1} \text{nm}^{-2}$. Moreover, most hexamer simulations for $K_{el} = [10,50,100]$ $\text{kJ mol}^{-1} \text{nm}^{-2}$ were not stable. Therefore, we conclude that both $K_{el} = 250 \text{ kJ mol}^{-1} \text{nm}^{-2}$ and $K_{el} = 500 \text{ kJ mol}^{-1} \text{nm}^{-2}$ are reasonable choices for the system at hand. As reported in the new Supplementary Figure 5g, h, angles computed for systems on the surface with these two strengths agree well with the experimental values (Figure 5a). Furthermore, there are sizeable populations of either bound or unbound spokes in all elastic constant configurations that have been investigated, as reported in the new Supplementary Figure 5i.

Other, minor remarks:

Martini force field version 2.6 is mentioned, but this version is non-existent (they may be confused with the version numbering of the martinize script?). Besides, no reference is given to the actual Martini protein force field that was used.

Thanks for pointing this out. The correct Martini force field is indeed version 2.2 -this has been corrected in the methods section.

It would be really helpful to include somewhere a Figure of the actual SAS-6 protein (dimer) structure.

Thank you for the useful suggestion. We have added a schematic of the SAS-6 protein homodimer (new Supplementary Figure 1c).

Reviewer #2 (Remarks to the Author):

In this paper Banterle et al., use highspeed AFM to examine the kinetics of Sas-6 ring assembly in vitro. The paper is quite complicated, but it is generally well written, the data is very novel and of high quality, and the authors come to several important and provocative conclusions. I think the paper will be of great interest to the centriole community, and also to those interested in understanding and modelling organelle assembly more generally. I am therefore strongly supportive of publication in Nature Communications, although I have a number of points the authors should consider prior to final acceptance.

We thank the reviewer for the positive feedback on our work and for the many important comments, which helped us improve the manuscript.

Major points:

1. Extrapolating the kinetics derived from the in vitro data described here to the situation in vivo is likely to be more complicated than the authors suggest. The authors estimate that in their in vitro experiments the surface shifts the equilibrium towards oligomerisation by a factor of $\sim 10^4$ (compared to solution) and then do a back-of-the-envelope calculation to show that if the same were true for the surface of a centriole, then the kinetics observed here would be sufficient to explain why daughter centrioles only form at the surface of the mother centriole. This calculation does not seem particularly useful to me. First, the 100 Sas-6 homodimers estimated to be present at human centrioles are probably mostly (usually entirely?) in the single stacked cartwheel of the

daughter centriole, not randomly spread across the entire centriole surface. Second, in the *in vitro* experiments, lots of cartwheels assemble on the surface, whereas *in vivo*, cartwheel assembly is restricted to a single site on the surface of the centriole, where it is thought that Plk4 promotes cartwheel assembly.

This does not in any way detract from the importance of making these *in vitro* measurements, but I think the authors should tell readers more forcefully that it is not necessarily straightforward to infer kinetics at the single site of cartwheel assembly *in vivo* based on these *in vitro* measurements with a single purified component.

We agree with the reviewer: extrapolating from the *in vitro* measurements to the *in vivo* situation should be considered merely at a coarse grain level, providing an order of magnitude estimate rather than an accurate numerical relationship. We have altered the corresponding text in the revised manuscript to clarify this point (page 9). See also response to reviewer 3, point 2.

2. In line 267 and line 275 the authors phrasing makes it sound as though it is very mysterious how a molecule that is intrinsically helical might form a closed ring when assembled on a surface. I may be missing something here, but to me this seems obvious, as the interaction with the flat surface is simply strong enough to force the ring to adopt a flat conformation.

The authors state that the observation of S-shaped conformations supports that idea that these closed rings would prefer to be helical, but in the movie I could see one such structure forming from the fusion of two open rings. If this can happen, does the existence of such structures still suggest that the structures prefer to be helical?

We apologize for the misunderstanding, which probably stemmed from excessive compaction of the text, making it difficult to comprehend the physical problem. Indeed, understanding why a structure that has an intrinsic helical propensity forms a closed ring when constrained on a surface is obvious. What is not obvious, however, is to determine the consequences of such a helical propensity on the properties of the ring assembled on the surface (e.g. calculating its radius from in-plane and out-of-plane angles of molecular bonds). Since the energy of the final assembly state does not depend on how that state is reached, neglecting dissipative forces, the energy of the ring assembled on the surface is the same as that of a hypothetical helix constrained on the surface. Calculating such energy for a pure helix has been addressed in the physics literature and is referred to as the squeelix theory (PMID: 27888445, PMID: 29498722). As mentioned in the manuscript, this theory predicts the existence of metastable S-shaped conformations on the surface, which we indeed observed by PORT-HS-AFM. Furthermore, the squeelix theory predicts that such transient states exist independently of how they are assembled, and the reviewer has indeed spotted an occurrence where merging of two higher order oligomers on the surface leads to a metastable S-shaped conformation. We expanded the explanation of this topic in the main text to hopefully clarify this point in a satisfactory manner (page 10). This is also highlighted in a new high magnification sequence visualizing the assembly of such an S-shaped configuration (Movie S3)

3. I found the last section of the results on how the asymmetry of Sas-6 ring polymers may guide ring-stacking a little confusing. I'm not an expert in molecular dynamic simulations so I did not fully understand the description of how this was performed nor the assumptions about the molecular interactions that are required to drive the apparent non-random orientation of the spokes in the closed rings. I assume there is some cooperativity, such that the angle of the spokes in one dimer influences the angles of the spokes in nearby dimers (although this can't be too strong or the ring would be trapped in an "all up" or "all down" orientation). But my real confusion was in trying to understand how the asymmetry of the rings formed in solution influences the assembly process to generate asymmetries in the stack—as the authors just told me that the assembly process is not driven by rings formed in solution but rather by oligomers assembling into rings on a flat surface. So, for example, the interactions shown in Fig.6C (between two undulating rings in solution) are not likely to occur on the flat surface.

I guess my confusion in part comes from not understanding how ring stacking occurs in vitro. Does a ring form on the surface and then oligomers assemble on top of this (using the tail of the spokes to guide assembly, which seems unlikely)? Or, are the authors arguing that rings can actually form in solution as well, and these can then stack on top of the surface rings (perhaps as depicted in Fig.6C).

We apologize for not having been sufficiently clear here. The reviewer is correct in thinking that the assembly process is not driven by rings forming in solution. The MD simulations of higher order oligomers in solution merely explore what would be their preferred intrinsic conformation. Albeit not necessarily reflecting the configuration they would adopt on a surface, such an analysis reveals the intrinsic asymmetries of such rings, which we propose could conceivably be maintained in a stack of rings assembled on the surface, and thus impart chirality. We clarified the reasoning in the main text of the revised manuscript (page 12). In addition, we now spell out explicitly that the model pertains to the end state and does not inform us on the dynamics or the polarity of the stacking reaction (page 12).

In addition, the authors state that “to examine ring-stacking in the cellular context” (line 318) they looked at in vitro assembled Sas-6 stacks. I don’t think the authors can be certain that the mechanisms of stacking in vitro necessarily reflect those in vivo, so again they should be more cautious in how they describe this.

Fully agreed. The text has been changed accordingly (page 12).

Minor Points:

1. The authors should explain in the main text that the Sas-6 protein they use here is a truncation, rather than referring to a previous publication. This is an important point that readers should be fully aware of.

We agree and now explicitly state this fact in the main text (page 6).

2. The authors nicely show that Sas-6 self-assembly is reversible in vitro. I think Cottee et al., (eLife, 2015) inferred that this must also be the case in flies in vivo (as a mix of WT and head-domain mutant forms of Sas-6 did not dominantly kill cartwheel assembly), which might be worth mentioning as it supports the authors in vitro conclusions.

We thank the reviewer for pointing this out. We now refer explicitly to this important publication in the revised manuscript (page 15).

3. Supp Fig. 3. In h) the authors say Fig.3b but I think mean Fig.3c.

We apologize for the mistake, which has been corrected (page 8 of the Supplementary material).

4. The red line and angle shown in Fig.S4d is not explained in the legend and is buried in the Methods.

We updated the description of the legend accordingly (page 10 of the Supplementary material).

Reviewer #3 (Remarks to the Author):

The authors present an elegant AFM work in which they were able to directly visualize the self-assembly of SAS-6, a key protein of the centriole organelle by the use of the high-speed (HS) time lapse AFM instrument. Importantly, they utilized the instrument termed PORT-HS-AFM capable of gentle touch of the sample. Such an instrumentation was a critical factor for reliable observation of the SAS-6 protein nanoscale dynamics, which includes association and dissociation steps along the self-assembly process. The authors acquired a dataset of SAS-6 oligomers as large as ~40, 000 species, which allowed them to quantitatively analyze the data using the oligomers assembly-disassembly model developed in the paper. Given that HS-AFM is a single molecule technique, such statistics is really impressive. Overall, the observations led the authors to a model of the self-

assembly of SAS-6 oligomers with ring shape explaining their assembly in the centriole cartwheel. The experimental data are supplemented by computer modeling of the SAS-6 oligomerization, which provided additional details to the three-dimensional shape of SAS-6 rings. My few comments to the paper are listed below.

We thank the reviewer for her/his appreciation of our work and for the valuable comments.

1. The time-lapse results demonstrate that SAS-6 protein have a high affinity to the mica surface, so the protein rather rapidly covers the surface. In parallel, the protein assembles in oligomers; however, their analysis of the kinetics of oligomers assembly does not include the contribution to the oligomers assembly from the protein species coming to bound oligomers from the bulk solution. Such events can be seen in the movie file. If the 2D diffusion is the predominant pathway for the oligomer's assembly, it should be justified.

The reasons for considering exclusively 2D diffusion as the path towards higher order oligomerization were indeed not sufficiently justified in the initial submission –apologies. The main reason for not considering the contribution of higher oligomeric species in solution is that the K_d for the head-to-head interaction that mediates such higher order oligomerization ($\sim 60 \mu\text{M}$, PMID: 21277013) is three orders of magnitude lower than the concentration employed here (120nM). Therefore, essentially no higher order oligomers should assemble in solution in these experiments, something that we confirmed since such oligomers were never observed landing onto the surface, as can be seen for instance in the initial frames of Supplementary Movie 1. That this is the case is now mentioned explicitly in the revised manuscript (page 6).

One could also imagine that homodimers present in solution would incorporate directly into higher order oligomers already present on the surface. To test whether this would have an impact on the system kinetics, we sought to fit the experimental dataset with a model explicitly considering such direct incorporation of homodimers from solution onto assembled oligomers. We found that the corresponding fitted rate was close to zero ($-1.81 \cdot 10^{-5} [\text{mol}/(\text{s} \cdot \mu\text{m}^3)]$), demonstrating that such a process is either negligible or absent. Prompted by the reviewer's comment, we wondered also whether SAS-6 monomers might be present in solution, which could conceivably impact the analysis. To address this possibility, we performed Circular Dichroism (CD) measurement of CrSAS-6[NL] and determined the monomer-dimer equilibrium constant to be $K_d = 20 \pm 15 \text{nM}$. Given that the monomer concentration in these experiments is 240 nM (120 nM for the homodimer), it follows that only a minute fraction of the protein is expected to be monomeric, thus resulting in a negligible impact on the analysis. This new CD data is reported in Supplementary Figure 1 d, e.

2. The role of the mica surface on the shape of SAS-6 rings needs to be discussed thoroughly. Additionally, mica is not a natural environment for cells, therefore a translation of the geometrical data for the rings assembled on this surface to their shape in natural environment such as surface of torus sounds quite speculative. Two factors, the surface shape and composition can contribute to the self-assembly process and the ring shape and these factors should be discussed.

Indeed, mica is not a natural environment for cells and the flat surface employed here has a distinct geometry from the toroidal surface of relevance in cells. As requested by the reviewer, we discuss these differences and limitations explicitly in the revised manuscript (pages 9). See also response to reviewer 2, major point 1.

3. The asymmetry of SAS-6 rings and their assembly in cartwheel is described along with the Cryo-EM results. However, these are data published by others, so it is appropriate to move this section of the paper to the discussion section.

We respectfully disagree with the reviewer on this point. Although the raw Cryo-EM data were published previously, as mentioned already explicitly in the initial submission, the processing of the data and the resulting interpretation are both novel results, so that it seems most logical to leave them in the namesake section.

Reviewer #4 (Remarks to the Author):

Centriole duplication occurs proximally to a parent centriole with the daughter centriole elongating orthogonally and remaining attached to the parent until the end of cell division. SAS-6 self-assembly results in a cartwheel-shaped feature that is one of the initial steps in the centriole duplication process. The prevalent hypothesis regarding centriole duplication places the polo-like kinase 4 (Plk4) as the key player in the initiation of the duplication process and a recruiter/regulator for the self-assembly of SAS-6. However, over-expression of SAS-6 results in supernumerary centrioles, suggesting an alternative self-assembly pathway. Banterle et al. characterized the effect of surfaces on the self-assembly of SAS-6 using HS-AFM (high-speed AFM, off-resonance mode with photothermal actuation) combined with coarse-grained molecular dynamics (CG-MD) simulations. Additionally, they provide a theoretical framework for the self-assembly process.

Authors show that, upon binding to the mica surface, SAS-6 homodimers assume spoke and head type conformation, which are capable of further self-assembly, through the head regions, into rings with protruding spokes. Furthermore, their experiments revealed that the self-assembly process is very dynamic with multiple pathways for ring assembly, ring-opening, and dissociation/fragmentation of rings and partial rings. Authors then use the obtained data to establish a theoretical framework and determine the appropriate rate constants for the self-assembly of SAS-6. Additionally, they investigated the physical reason for the observed phenomenon using CG-MD simulations.

The key takeaways are:

- Ring formation efficiency exhibits a strong dependence on effective surface concentration.
- Ring formation is reversible.
- A coagulation-fragmentation model is proposed that supposes that all intermediates react with one another, which greatly facilitates the self-assembly due to faster reaction convergence.
- Surface effects ensure that the rings are nine-fold symmetric, not flat, and exhibit non-random torsional angle between homodimers; which has implications for the stacking of SAS-6 rings in the centriole duplication process and may explain its chirality.

Overall, the manuscript expands the current knowledge about SAS-6 self-assembly with a clear narrative. In addition to the scientific merit, the methodological value presented by the semi-automatic data-processing routines (using freely available software) is also of notice and commendation. Although of high quality, the manuscript has a few minor issues

We thank the reviewer for the appreciation of our work and for the helpful comments, which contributed to improving the manuscript.

- Authors should provide close-up movies for each type of assembly presented in Fig. 1. Similar to how the opening-closing is shown in Supp. Movie 2. Similarly, the S-shaped assembly should be presented in a separate movie.

This is a very nice suggestion, which we have followed by incorporating four additional supplementary movie insets now embedded into Supplementary Movie 1. Moreover, we provide another Supplementary Movie 3 with a S-shaped higher oligomeric species.

- Supp. Fig. 3a legend: Fig. 3b should be Fig. 3c. Although, the open and closed states have been added and the panel is not exactly the same. Similar internal refs. should be checked.

We apologize for the mistake, which has been corrected.

- References need a thorough editing as some are incorrect (probably due to manuscript iterations). E.g. Fig. 5b, ref. 5 should be ref. 8?

We apologize for the mistakes; all references have been checked and all issues have been resolved.

- An opinion: I strongly urge the authors to publish the MD results and corresponding analysis in Scientific Data (or similar journal). Availability of an extended version of the MD data and analysis may prove helpful for other researchers with other foci.

We thank the reviewer for encouraging us to make this data set readily accessible. As a result, the MD results are now deposited and publicly available at the open-access research data repository of Heidelberg University ([heidATA \(uni-heidelberg.de\)](https://www.uni-heidelberg.de/heidata)), as also stated in the Data Availability section (page 45).

Reviewer #5 (Remarks to the Author):

Summary

The authors investigate how SAS-6 assembles into ring polymers, which are an essential component of the centriole organelle.

The authors use high-speed atomic-force microscopy to observe the assembly process on a surface. They use machine learning to segment and classify the various intermediate states and of the polymer assembly.

They propose a system of differential equations to describe the assembly process and go on to determine the parameters of their model based on the observed frequencies (determined via ML classification) of the different intermediate states.

Based on their observations and simulations, the authors reason that the presence of a surface is essential for the assembly of ring polymers as they occur in the centriole and go on to develop a theory for the stacking behavior of polymer rings, which is validated via cryo-electron tomography.

Scope of this Review

I really only feel qualified to comment on the machine learning aspect of the paper. I regret that I cannot comment on the other aspects.

Machine Learning Segmentation and Classification

The authors use the Ilastic software for segmentation of their AFM images, as well as for classification of the identified objects.

Ilastic is a solid choice and by now a well established tool.

Image processing of the AFM images is done in two steps.

First the images are segmented via pixel classification, that is each pixel is assigned a probability of belonging to the background, head domain, or spokes.

Based on this pixels are grouped to objects, which are then again classified by Ilastic and assigned a probability of belonging to one of the oligomer types.

I appreciate it that the authors illustrate the process in Supplementary Figure 2.

While I absolutely agree with the image processing pipeline, there is also a little room for criticism: It looks like the classification in particular is a challenging problem with this data and far from trivial. The authors mention in the caption for Supplementary Figure 2 that they train the classifier by manually assigning "some oligomers with clearly identifiable spoke numbers" to the corresponding class. So it seems that at times the correct spoke number is not easily identifiable, even for a human expert. I believe Ilastic cannot learn to identify spoke numbers in ambiguous

oligomers without receiving properly annotated training data for exactly this ambiguous type of oligomers.

Can the authors comment on this problem?

Are there oligomers that are not identifiable by a human?

Roughly what fraction would fall into this category?

After classification by Ilastic are these what class are these oligomers assigned to?

Do they receive a high uncertainty label, i.e. a high entropy probability distribution?

How confident are the authors that their classification is reliable?

Note that the fact that Ilastic outputs a probability distribution, i.e. reflects the uncertainty of its decision does not necessarily mean that its classifications are correct.

It is for example possible that Ilastic assigns a wrong label to an object with a high confidence.

A standard way to address such concerns about the classification quality would be to perform validation on a part of the dataset that was not used during training.

That is, one could randomly choose a subset of objects that were not used during training and manually classify them.

Then these ground truth object labels can be compared to the Ilastic prediction to compute accuracy, confusion matrix, or other metrics.

I would appreciate it if such a validation was included in the final version of the paper, or if the authors would comment on the points mentioned above.

We thank the reviewer for bringing up these important points. There are indeed cases in which attribution of higher oligomerization state can be ambiguous for a human annotator. For the human eye, differences among head domains are difficult to detect; by contrast, the number of spokes is usually telling. However, there are some cases where the number of spokes cannot be counted with certainty, perhaps because they were motile during imaging. Therefore, we used solely higher order oligomers with clearly countable spokes to classify the training set. Importantly, whereas spokes are used by Ilastic at the segmentation stage, the classification is based exclusively on the recognition of head domains, so that uncertainties in spoke counts have no impact for the final output. That this is the case is now explained further in the Methods section (page 35).

To validate the accuracy of the automatic classification, we followed the reviewer's suggestion: two of the authors (NB, PG) performed manual classifications of all particles present in 12 frames across an entire movie, excluding those particles previously used for training. Given that the output of the automatic classification are probabilities and not one class with the highest probability, rather than computing the accuracy and confusion matrix, we compared the number of each oligomeric species determined by the automatic classification with that estimated by the two individuals. This analysis is reported in the new Supplementary Figure 2 g-t and shows that the difference between the automatic and the average manual classification is within the estimated error range for each oligomeric species. Therefore, automatic classification, albeit not perfect, is not worse than human classification. This point is also discussed explicitly in the revised manuscript (page 8).

- Reviewers' Comments:

Reviewer #1:

Remarks to the Author:

I am happy with the revisions made by the authors, and have no further points to consider.

Reviewer #2:

Remarks to the Author:

In their revised manuscript Banterle et al. have done a good job of addressing my initial concerns. The paper is still quite complicated, but I was able to much better understand the areas that I struggled with in the previous version. I am strongly supportive of publication.

The only minor point the authors might like to consider is the revised Figure S2g-t. This Figure has now changed and, to the non-expert, it looks as though the AI algorithm does not do a great job of identifying some of the structures (e.g. closed circles with 10 dimers) compared to the manual annotators. Moreover, the description in the legend that >95% of the data points fall within 2 sigmas of the automated classification doesn't mean much to me (Is that fantastic or only OK? Can you translate this into a more useful approximate error rate for each of the classes?). It might be worth expanding on the description of this data, as it is central to the manuscript that this automated scoring is accurate.

Reviewer #3:

Remarks to the Author:

The authors addressed my critiques.

Reviewer #4:

Remarks to the Author:

Authors have addressed the issues identified in the previous round of review.

Reviewer #5:

Remarks to the Author:

I very much appreciate the additional manual verification of the classification accuracy done by the authors.

I am happy with the proposed changes and do not see a problem with publication, considering the machine learning aspect of the paper.